# FOCUS: EFFICIENT KEYFRAME SELECTION FOR LONG VIDEO UNDERSTANDING

**Zirui Zhu**[1,2][†]    **Hailun Xu**[2]    **Yang Luo**[1]    **Yong Liu**[1]
**Kanchan Sarkar**[2][◇][†]    **Zhenheng Yang**[2][◇][†]    **Yang You**[1][†]
[1]National University of Singapore    [2]TikTok
[◇] Project Lead    [†] Corresponding Author
{zirui, youy}@comp.nus.edu.sg
{kanchan.sarkar, yangzhenheng}@tiktok.com

## ABSTRACT

Multimodal large language models (MLLMs) represent images and video frames as visual tokens. Scaling from single images to hour-long videos, however, inflates the token budget far beyond practical limits. Popular pipelines therefore either uniformly subsample or apply keyframe selection with retrieval-style scoring using smaller vision-language models. However, these keyframe selection methods still rely on pre-filtering before selection to reduce the inference cost and can miss the most informative moments.

We propose FOCUS, *Frame-Optimistic Confidence Upper-bound Selection*, a training-free, model-agnostic keyframe selection module that selects query-relevant frames under a strict token budget. FOCUS formulates keyframe selection as a combinatorial pure-exploration (CPE) problem in multi-armed bandits: it treats short temporal clips as arms, and uses empirical means and Bernstein confidence radius to identify informative regions while preserving exploration of uncertain areas. The resulting two-stage exploration-exploitation procedure reduces from a sequential policy with theoretical guarantees, first identifying high-value temporal regions, then selecting top-scoring frames within each region. Extensive experiments across four long-video question-answering benchmarks and four popular MLLMs demonstrate that FOCUS delivers substantial accuracy improvements while processing less than 2% of video frames. For videos longer than 20 minutes, it achieves an 11.9% gain in accuracy on LongVideoBench, demonstrating its effectiveness as a keyframe selection method and providing a simple and general solution for scalable long-video understanding with MLLMs.

## 1 INTRODUCTION

> *"The art of being wise is the art of knowing what to overlook."* — William James

Recent advances in large language models (LLMs) and multimodal large language models (MLLMs) have significantly improved visual understanding and reasoning. In current frameworks, images are encoded into visual tokens aligned with text and jointly processed by the LLM. Extending this paradigm to videos—especially long, untrimmed ones—introduces a key challenge: the sheer number of frames leads to an overwhelming number of visual tokens, making inference computationally prohibitive.

A common solution is aggressive downsampling (Wang et al., 2022b; Lin et al., 2023; Maaz et al., 2024; Zhang et al., 2025c), but uniformly sampling a handful of frames (e.g., 64 from a one-hour video) often misses critical content (Tang et al., 2025; Zhang et al., 2025b). Increasing the frame rate, on the other hand, causes token explosion (Wang et al., 2024c). This trade-off motivates the need for keyframe selection: choosing a small set of informative frames that preserve semantics while staying within token limits.

Recent methods address this by scoring frame relevance with pre-trained vision-language encoders (e.g., CLIP (Radford et al., 2021) or BLIP (Li et al., 2022)) and then pick the highest-relevance

frames (Tang et al., 2025; Zhang et al., 2025b). These text-image matching approaches are typically training-free and plug in easily before the visual encoder in MLLM stacks, retrieving frames with higher relevance other than uniform sampling. Despite their success, current keyframe selection methods still face scalability and efficiency limitations. For a one-hour video at 30 fps (over $10^5$ frames), exhaustively scoring all frames entails on the order of $10^{11}$-$10^{12}$ FLOPs with a vision-language encoder like BLIP (Li et al., 2022). This scaling pressure forces existing methods to uniformly sample the video to lower frame rate before the scoring process. This pre-filtering process before keyframe selection undermines the goal of identifying most informative keyframes from all frames (Zhang et al., 2025b; Tang et al., 2025).

In this work, we propose FOCUS, *Frame-Optimal Confidence Upper-Bound Selection*, a training-free, plug-and-play keyframe selection method designed to process extremely long videos with minimal computational overhead. FOCUS is easy to implement in practice while offering an elegant theoretical foundation.

The key insight behind FOCUS is grounded in the observation that natural videos exhibit strong temporal locality: adjacent frames are highly correlated in appearance and motion (Wiegand et al., 2003; Wang et al., 2016; 2022b). This local smoothness naturally extends to frame-query relevance scores.

Concretely, for each video-query pair we compute a frame-level relevance sequence $\{r_t\}$, where $r_t$ is the cosine similarity between the visual embedding of frame $t$ and the text embedding of the query produced by BLIP. We then measure temporal dependence via the autocorrelation function (ACF) $\rho(\delta) = \mathrm{corr}(r_t, r_{t+\delta})$ at lag $\delta$ (in seconds), and aggregate $\rho(\delta)$ across videos. As illustrated in Figure 1, both LongVideoBench and Video-MME exhibit strong short-range correlation: the median ACF remains above $0.5$ for roughly the first 5 seconds.

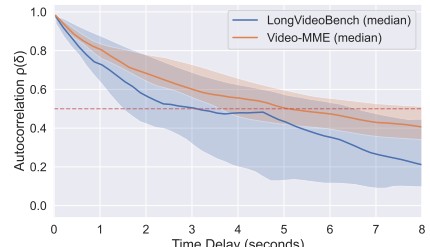

Figure 1: Temporal autocorrelation (ACF) of per-frame query relevance on LongVideoBench and Video-MME. We compute frame-level relevance per video and take the ACF over time lags (seconds); solid lines show the median across videos and shaded bands the interquartile range. The dashed line marks the correlation half-life level ($\rho(\delta) = 0.5$).

This observation implies that exhaustive scoring of all frames is unnecessary. Instead, we can formulate keyframe selection as a bandit problem to adaptively allocate computation: quickly filtering out irrelevant temporal regions, concentrating scoring on promising segments, and ultimately prioritizing the most informative keyframes.

FOCUS first partitions the video into short temporal clips, each treated as an arm in a multi-armed bandit. The clip selection is then framed as a Combinatorial Pure-Exploration (CPE) problem: the goal is to identify a subset of arms that maximizes expected cumulative relevance under a limited budget. Each arm maintains an empirical mean relevance and a Bernstein-style confidence radius. Computation is adaptively allocated to clips that are either promising (high mean) or uncertain (large confidence radius), following an optimism-in-the-face-of-uncertainty principle. This iterative process enjoys theoretical convergence guarantees. To leverage parallel computation, we reduce the iterative strategy to a coarse-to-fine schedule: optimistic means guide exploration, while unbiased empirical means inform final arm selection. Within each selected arm, we extract the top-relevance frames to construct the final keyframe set.

We validate the effectiveness of our approach on two video understanding benchmarks, including LongVideoBench (Wu et al., 2024) and Video-MME (Fu et al., 2025). The proposed FOCUS is tested as an off-the-shelf module on with four popular MLLMs. FOCUS improves answer accuracy over state-of-the-art keyframe selection baselines across benchmarks while maintaining lower inference cost. The gains are especially pronounced on long-form videos: for videos longer than 20 minutes on LongVideoBench, FOCUS delivers a 11.9% accuracy improvement while still cutting inference cost.

In summary, our main contributions are three-fold: (1) We formulate query-aware keyframe selection as a budgeted *combinatorial pure-exploration* (CPE) problem in a multi-armed bandit setting; (2) We introduce FOCUS, a training-free, model-agnostic keyframe selection module that selects query-

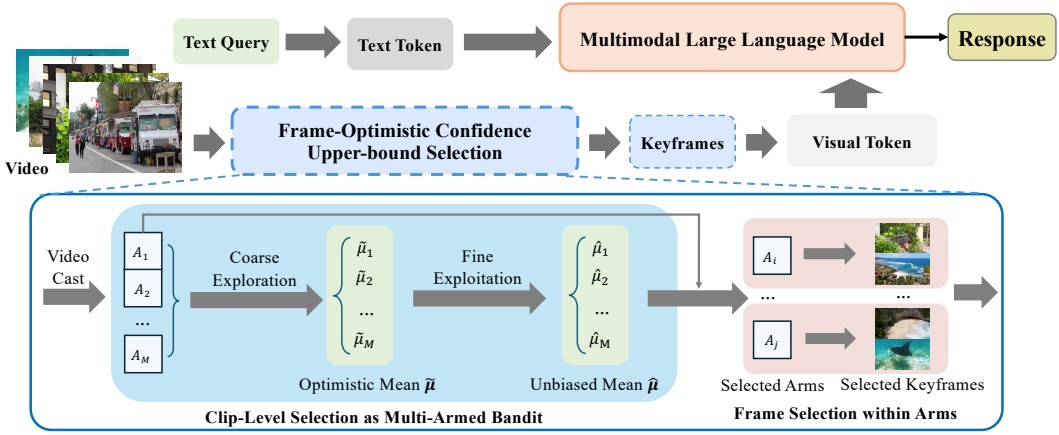

Figure 2: Overview of FOCUS. FOCUS partitions videos into fixed-length clips as bandit arms, applies optimistic confidence upper-bound arm selection and selects final keyframes within each promising arms.

relevant frames under a strict token budget; (3) We validate the effectiveness of FOCUS on two long-video understanding benchmarks, achieving consistent gains across four popular MLLMs.

## 2 METHOD

### 2.1 PROBLEM FORMULATION

**Keyframe Selection Setup.** Let a video be $V = (\boldsymbol{x}_1, \ldots, \boldsymbol{x}_T)$ and denote the corresponding text query as $q$. Let the frame index set be $\mathbb{T} = \{1, \ldots, T\}$. A downstream MLLM $\Phi$ consumes a subset of frames indexed by $\mathbb{K} \subseteq \mathbb{T}$ with $|\mathbb{K}| = k$ and produces an answer $\hat{a} = \Phi\big(q, \{\boldsymbol{x}_t\}_{t \in \mathbb{K}}\big)$. Let $R_\Phi(\mathbb{K} \mid V, q)$ denote the task-level utility of the selected frames (e.g., quality of generated answer, relevance to query, or other performance metrics).

**Oracle and Surrogate Objective.** The oracle objective chooses $\mathbb{K}$ to maximize expected utility:

$$\mathbb{K}^{\text{oracle}}(V, q) = \underset{\mathbb{K} \subseteq \mathbb{T},\, |\mathbb{K}| = k}{\arg\max}\; \mathbb{E}\big[R_\Phi(\mathbb{K} \mid V, q)\big], \tag{1}$$

Direct optimization to equation 1 is infeasible due to the combinatorial search space and the high cost of black-box evaluations of $\Phi$. We further expand the task-level utility $R_\Phi(\mathbb{K} \mid V, q)$ to a summation of frame-level utility $y_t \in [0, 1]$:

$$\mathbb{K}^\star = \underset{\mathbb{K} \subseteq \mathbb{T},\, |\mathbb{K}| = k}{\arg\max}\; \mathbb{E}\Big[\sum_{t \in \mathbb{K}} y_t\Big]. \tag{2}$$

However, estimating the contribution of each frame $t$ to the task-level utility is also intractable. We therefore posit that $y_t$ is indirectly observable via a vision-language encoder $\psi$ that outputs a relevance score $r_t = \psi(\boldsymbol{x}_t, q; \boldsymbol{\theta}) = y_t + \epsilon_\psi$, where $\epsilon_\psi$ denotes encoder-induced noise. We assume $\epsilon_\psi$ follows some distribution that are supported on $[0, 1]$ and with zero mean and $\sigma_\psi^2$ variance. Under this assumption, the relevance score $r_t$ is a unbiased estimator of $y_t$ which is also commonly used in many works (Tang et al., 2025; Yu et al., 2024) implicitly.

Exhaustively scoring all $T$ frames to get $\{r_t\}$ is computationally prohibitive, especially for hourly long videos which contains over $10^5$ frames. This computational constraint motivates us to model keyframe selection under budget constraints, where we strategically allocate a limited sampling budget to identify the most promising temporal segments before producing the final set of $k$ keyframes. Instead of directly optimizing equation 2 at the frame level, we will approximate it through a combinatorial pure-exploration multi-armed bandit formulation at the clip level, which significantly reduces exploration cost.

---

**Algorithm 1** Iterative Optimistic Confidence Upper-bound Arm Selection

---

**Require:** Maximization oracle $\text{TopM}(\{\mu_a\}, m) \to \mathbb{A} \subseteq \mathcal{A}$
 1: **Initialize:** Empirical means $\hat{\mu}_0(a) \leftarrow 0$ and $N_0(a) \leftarrow 0$ for all $a$.
 2: Pull each arm $a \in \mathcal{A}$ for $q$ times and observe the rewards.
 3: $n \leftarrow mq$ and $N_a(n) \leftarrow q$ for all $a$.
 4: Update empirical means $\hat{\mu}_a(n)$ for all $a$.
 5: **for** $n \leftarrow mq, mq+1, \ldots$ **do**
 6:     $\mathbb{A}_n \leftarrow \text{TopM}(\hat{\boldsymbol{\mu}}, m)$
 7:     Compute confidence radius $\beta_a(n)$ for all $a \in \mathcal{A}$          $\triangleright \beta_a(n)$ defined in equation 5
 8:     **for** $a \leftarrow 1$ **to** $M$ **do**
 9:         **if** $a \in \mathbb{A}_n$ **then**
10:             $\tilde{\mu}_a(n) \leftarrow \hat{\mu}_a(n) - \beta_a(n)$
11:         **else**
12:             $\tilde{\mu}_a(n) \leftarrow \hat{\mu}_a(n) + \beta_a(n)$
13:         **end if**
14:     **end for**
15:     $\tilde{\mathbb{A}}_n \leftarrow \text{TopM}(\tilde{\boldsymbol{\mu}}, m)$
16:     **if** $\tilde{\mathbb{A}}_n = \mathbb{A}_n$ **then**
17:         **return** $\mathbb{A}_n$
18:     **end if**
19:     $p_n \leftarrow \underset{a \in (\tilde{\mathbb{A}}_n \setminus \mathbb{A}_n) \cup (\mathbb{A}_n \setminus \tilde{\mathbb{A}}_n)}{\arg\max} \beta_a(n)$          $\triangleright$ break ties arbitrarily
20:     Pull arm $p_n$ and observe the reward
21:     Update empirical means $\hat{\mu}(p_n)$ with the observed reward
22:     $N_{p_n}(n+1) \leftarrow N_{p_n}(n) + 1$
23: **end for**

---

## 2.2 Clip-Level Selection as Multi-Armed Bandit

For a video $V = (\boldsymbol{x}_1, \ldots, \boldsymbol{x}_T)$, we partition the timeline into $M$ non-overlapping fixed-length clips $\mathcal{A} = \{A_a\}_{a=1}^M$, where each clip $A_a \subseteq \mathbb{T}$ spans frames $[s_a, e_a]$ and is treated as a bandit arm. We define pulling arm $a$ as uniformly sampling a frame $t \in A_a$ and observing its query relevance score $r_t$ as the reward. The unseen frame-level utility of the sampled frame is modeled as $y_t \sim \nu_a$, where $\nu_a$ has mean $\mu_a$ and variance $\sigma_a^2$.

Intuitively, our goal is to focus on the most promising clips which means we have to identify the optimal subset $S^\star \subseteq \mathcal{A}$. Formally, we define the *decision class* $\mathbb{S} \in 2^{\mathcal{A}}$ as a subset of the power set of $\mathcal{A}$. The optimal member $S^\star$ of decision class $\mathbb{S}$ is defined as

$$S^\star = \arg\max_{S \in \mathbb{S}} \sum_{a \in S} \mu_a. \tag{3}$$

Under the classic CPE framework, the learner's objective is to identify $S^\star$ after interacting with the arms over a sequence of rounds. In the keyframe selection setting, our final goal is to further select $k$ keyframes from the selected arms. Denote $\{k_a\}_{a=1}^{|S^\star|}$ as the number of keyframes allocated to the $a$-th selected arm. We further define the frame-level optimal keyframe subset $\mathbb{K}_a^\star$ as

$$\mathbb{K}_a^\star = \arg\max_{\mathbb{K}_a \subseteq A_a, \, |\mathbb{K}_a| = k_a} \sum_{t \in \mathbb{K}_a} y_t. \tag{4}$$

The final keyframe subset $\mathbb{K}^\star$ is then defined as $\mathbb{K}^\star = \bigcup_{a \in S^\star} \mathbb{K}_a^\star$. Empirically, we assume the decision class $\mathbb{S}$ is all size-$m$ subsets of $\mathcal{A}$ and keyframes are equally distributed across the promising arms. This setting gives us an elegant theoretical guarantee of regret bound as shown in section C and is also proved to be effective in our experiments.

## 2.3 Optimistic Confidence Upper-bound Arm Selection

### 2.3.1 Optimal Arm Selection.

Generally, we play a exploration game by pulling an arm $a$ and observing the reward $r_t$ at each round $n$. We maintain two core empirical statistics for each arm $a$ during this process: an empirical mean

---

**Algorithm 2** FOCUS: Frame-Optimistic Confidence Upper-bound Selection

---

**Require:** Maximization oracle $\text{TopM}(\{\mu_a\}, m) \rightarrow \mathbb{A} \subseteq \mathcal{A}$
 1: **Initialize:** Empirical means $\hat{\mu}_0(a) \leftarrow 0$ and $N_0(a) \leftarrow 0$ for all $a$.
     *// Stage I: Coarse exploration*
 2: Pull each arm $a \in \mathcal{A}$ for $q$ times and observe the rewards.
 3: $n \leftarrow mq$ and $N_a(n) \leftarrow q$ for all $a$.
 4: Update empirical means $\hat{\mu}$ for all $a$.
 5: Compute confidence radius $\beta_a(n)$ for all $a \in \mathcal{A}$
 6: $\tilde{\mu}_a(n) \leftarrow \hat{\mu}_a(n) + \beta_a(n)$ for all $a \in \mathcal{A}$
 7: $\mathbb{A}_{\text{coarse}} \leftarrow \text{TopM}(\tilde{\boldsymbol{\mu}}, m)$                    ▷ Optimistic Means UCB
     *// Stage II: Fine-grained exploitation*
 8: Pull each arm $a \in \mathbb{A}_{\text{coarse}}$ for $z$ times and observe the rewards.
 9: Update empirical means $\hat{\mu}_a(n)$ for $a \in \mathbb{A}_{\text{coarse}}$
10: $\mathbb{A}_{\text{fine}} \leftarrow \text{TopM}(\hat{\boldsymbol{\mu}}, m)$                       ▷ Unbiased Empirical Means
11: **return** $\mathbb{A}_{\text{fine}}$

---

$\hat{\mu}_a(n)$ and an empirical Bernstein confidence radius (variance-adaptive) $\beta_a(n)$, following the UCV-V style bound (Audibert et al., 2009):

$$\beta_a(n) = \sqrt{\frac{2\,\hat{\sigma}_a^2\,\ln n}{\max(1, N_a(n))}} + \frac{3\ln n}{\max(1, N_a(n))}. \tag{5}$$

Here $\hat{\sigma}_a^2$ is the empirical variance of arm $a$, $N_a(n)$ is the number of pulls for arm $a$ at round $n$ and $n = \sum_{a \in \mathcal{A}} N_a(n)$ is the total number of pulls. The confidence radius ensures that the empirical mean is within the confidence radius of the true mean with high probability, *i.e.,*

$$\mathcal{P}\left[|\hat{\mu}_a(n) - \mu_a| \leq \beta_a(n)\right] \geq 1 - \frac{6}{n}. \tag{6}$$

Please refer to Appendix B for the detailed proof.

As shown in Algorithm 1, the optimistic confidence upper-bound arm selection starts with an initialization phase where we pull each arm for $q$ times and observe the relevance scores as rewards. We then update the empirical means $\hat{\mu}_a$ and compute the confidence radius $\beta_a(n)$ for each arm $a$. Note the relevance score $r_t$ is an unbiased estimator of $y_t$ so we have $\mathbb{E}[\hat{\mu}_a] = \mu_a$. Then we choose the best $m$ arms using the empirical means $\hat{\mu}_a(n)$, *i.e.,* $\mathbb{A}_n = \text{TopM}(\hat{\boldsymbol{\mu}}, m)$, where $\hat{\boldsymbol{\mu}}$ is the vector of all arms' empirical means and $\text{TopM}(\cdot, m)$ returns a set of the $m$ arms with the largest empirical means.

We further refine the arm selection by evaluating the "potential" of each arm. To be specific, for arm $a \in \mathbb{A}_n$, we compute the lower confidence bound of the empirical mean, *i.e.,* $\text{LCB}_a(n) = \hat{\mu}_a(n) - \beta_a(n)$; for arm $a \notin \mathbb{A}_n$, we compute the upper confidence bound of the empirical mean, *i.e.,* $\text{UCB}_a(n) = \hat{\mu}_a(n) + \beta_a(n)$. If

$$\max_{a \notin \mathbb{A}_n} \text{UCB}_a(n) \geq \min_{a \in \mathbb{A}_n} \text{LCB}_a(n), \tag{7}$$

this indicates that some arms outside the current top-$m$ set are still potential to be included in the top-$m$ set. Thus, we choose the arm $a$ that we are most uncertain about, *i.e.,*

$$a = \underset{a \in (\tilde{\mathbb{A}}_n \setminus \mathbb{A}_n) \cup (\mathbb{A}_n \setminus \tilde{\mathbb{A}}_n)}{\arg\max} \beta_a(n). \tag{8}$$

We then pull this arm $a$ for $q$ times and repeat the process until the top-$m$ set is unchanged, *i.e.,* $\mathbb{A}_{n+1} = \mathbb{A}_n$. We then return the top-$m$ set $\mathbb{A}_n$.

It is easy to see Algorithm 1 is guaranteed to return the optimal top-$m$ set $\mathbb{A}_n$ with high probability (see Section C for the detailed proof). However, the iterative process is empirically inefficient (or intractable) as the sequential arm-pulls and updating can not be parallelizable. We have to pull the arms one-by-one which means forward the vision-language model with batch size 1 sequentially. This costs significant waste of GPU utilization.

### 2.3.2 TWO-STAGE ARM SELECTION.

To make the procedure practical and easy to parallelize, we specialize Algorithm 1 into the two-stage, batch variant in Algorithm 2. The overall framework is shown in Figure 2.

**Stage I: Coarse initialization.** We pull each arm $q$ times in parallel and update the empirical means $\hat{\mu}_a$ and confidence radii $\beta_a(n)$ for all $a \in \mathcal{A}$. This stage coincides with the initialization phase of Algorithm 1 and serves as a coarse exploration pass that produces reliable per-arm statistics at low coordination cost.

**Stage II: Fine-grained exploration (batched).** Using the optimistic scores $\tilde{\mu}_a(n) = \hat{\mu}_a(n) + \beta_a(n)$, we select the top $\alpha m$ arms, $\mathcal{A}_{\text{coarse}} = \text{TopM}(\tilde{\boldsymbol{\mu}}, , \alpha m)$, and allocate an additional $z$ pulls to each $a \in \mathcal{A}_{\text{coarse}}$ (performed in a single batch). Here, $\alpha$ is a hyperparameter that controls the ratio of the coarse exploration budget to the fine-grained exploration budget. This stage is a batched counterpart of the iterative loop in Algorithm 1: it implements the "optimism in the face of uncertainty" principle by concentrating samples on arms with the largest UCB values, while avoiding per-step scheduling overhead.

**Final Arm Selection.** After the fine exploitation, we form the final set by selecting the best $m$ arms according to the unbiased empirical means, $\mathbb{A}_{\text{fine}} = \text{TopM}(\hat{\boldsymbol{\mu}}, m)$. This choice mirrors $\delta$-PAC identification routines, where optimistic scores guide exploration but the recommendation itself is based on the empirical means $\hat{\mu}_a(n)$ rather than the optimistic means $\tilde{\mu}_a(n)$.

### 2.4 FRAME SELECTION WITHIN SELECTED ARMS

Given the selected arm set $\mathbb{A}_{\text{fine}}$ and a total budget of $K$ frames, we sample $k_a$ frames per arm $a \in \mathbb{A}_{\text{fine}}$ with equal allocation (i.e., $k_a = \text{round}(k/|\mathbb{A}_{\text{fine}}|)$, adjusted to sum to $K$). For each arm $a$ with index set $\mathbb{T}_a$ and observed rewards $\{r_{a,s}\}_{s \in S_a}$ at sampled indices $T_a \subseteq \mathbb{T}_a$, we simply interpolate all rewards $\hat{r}_{a,t}$ within the arm using the nearest-neighbor assignment. We then form a per-arm sampling distribution according to the interpolated rewards and draw $k_a$ frames *without replacement* from $p_a$. The final keyframe set is $\mathcal{K} = \bigcup_{a \in \mathcal{A}_{\text{fine}}} \mathcal{K}_a$.

## 3 EXPERIMENTS

### 3.1 EXPERIMENTAL SETUP

**Benchmarks** We follow the LMMs-Eval framework Zhang et al. (2024a) and adopt the open-source evaluation protocol from AKS for benchmarks, prompts, and scoring. Our experiments focus on two long-video multiple-choice QA benchmarks: LongVideoBench Wu et al. (2024) and VideoMME Fu et al. (2025). These datasets feature videos lasting up to an hour, where effective keyframe selection becomes crucial for performance. To ensure fair comparison (Tang et al., 2025), we disable subtitles, perform zero-shot evaluation, and keep model parameters frozen—varying only the frame selection strategy (our method versus uniform sampling). We also evaluate on MLVU (Zhou et al., 2025) and VSI-Bench (Yang et al., 2025) to assess generalization; detailed results on MLVU and VSI-Bench are provided in Section F.

**Implementation Details** We test both open-source video MLLMs (Qwen2VL (Wang et al., 2024a), LLaVA-OV (Li et al., 2025), LLaVA-Video (Zhang et al., 2025c) and Qwen2-7B (Yang et al., 2024) language model) and the commercial GPT-4o (0513). For frame relevance scoring, we use BLIP ITM (Li et al., 2022) to compute $r_t = \psi(\boldsymbol{x}_t, q; \boldsymbol{\theta})$, where $r_t$ estimates the latent frame-level utility as described in Section 2.1, which is justified as a promising choice by Tang et al. (2025). This also ensure a fair comparison setting as the frame-level utility is estimated using the same model.

### 3.2 PERFORMANCE ANALYSIS

We evaluate FOCUS by using it to select keyframes as the visual input for the four aforementioned MLLMs, and compare it against the commonly used uniform sampling strategy. The results on LongVideoBench and Video-MME are summarized in Table 1.

| Model | #Frame | LLM | LongVideoBench | Video-MME |
|---|---|---|---|---|
| GPT-4V | 256 | – | 61.3 | 59.9 |
| Gemini-1.5-Flash | 256 | – | 61.6 | 70.3 |
| Gemini-1.5-Pro | 256 | – | 64.0 | 75.0 |
| VideoLLaVA | 8 | 7B | 39.1 | 39.9 |
| MiniCPM-V 2.6 | 64 | 8B | 54.9 | 60.9 |
| InternVL2-40B | 16 | 40B | 59.7 | 61.2 |
| LLaVA-Video-72B | 64 | 72B | 63.9 | 70.6 |
| GPT-4o | 32 | – | 51.6 | 61.8 |
| GPT-4o *w/* Ours | 32 | – | **54.8** ↑ 3.2 | **62.5** ↑ 0.7 |
| Qwen2-VL-7B | 32 | 7B | 55.6 | 57.4 |
| Qwen2-VL-7B *w/* Ours | 32 | 7B | **62.3** ↑ 6.7 | **59.7** ↑ 2.3 |
| LLaVA-OV-7B | 32 | 7B | 54.8 | 56.5 |
| LLaVA-OV-7B *w/* Ours | 32 | 7B | **60.7** ↑ 5.9 | **58.3** ↑ 1.8 |
| LLaVA-Video-7B | 64 | 7B | 58.9 | 64.4 |
| LLaVA-Video-7B *w/* Ours | 64 | 7B | **63.5** ↑ 4.6 | **65.4** ↑ 1.0 |

Table 1: Video-question answering accuracy (%) of various MLLMs on LongVideoBench and Video-MME. FOCUS is integrated into GPT-4o, Qwen2-VL, LLaVA-OV, and LLaVA-Video. The suffix "*w/* Ours" denotes models using keyframes selected by our method; otherwise, frames are uniformly sampled. **#Frame** indicates the number of frames provided to the MLLM, and **LLM** denotes the language model size. We also include performance of additional popular MLLMs for reference.

**Improved Performance via Frame Selection.** As shown in Table 1, FOCUS consistently outperforms uniform sampling across both open-source and closed-source MLLMs on both LongVideoBench and Video-MME.

Specifically, on LongVideoBench, FOCUS improves accuracy by 3.2% on GPT-4o, 6.7% on Qwen2-VL-7B, 5.9% on LLaVA-OV-7B, and 4.6% on LLaVA-Video-7B. On Video-MME, the gains are 0.7%, 2.1%, 1.8%, and 1.0% on the same models, respectively.

We observe a clear trend that larger MLLMs with more frame inputs tend to achieve better performance. However, FOCUS significantly narrows this gap by identifying the most informative frames, thereby boosting the performance of smaller MLLMs. For instance, Qwen2-VL-7B with FOCUS outperforms Gemini-1.5-Flash on LongVideoBench, despite using 8× fewer input frames. This highlights the effectiveness of FOCUS as a plug-and-play keyframe selection module for a wide range of MLLMs.

**Interpretability through Visualizations.** We visualize the frames selected by FOCUS alongside uniformly sampled frames for two examples from LongVideoBench and Video-MME in Figure 3.

Note that LongVideoBench and Video-MME differ substantially in how their video-question pairs are constructed. In general, LongVideoBench features more detailed and specific questions, while Video-MME focuses on concise, high-level queries. Moreover, LongVideoBench tends to ask about specific scenes or events, whereas Video-MME emphasizes global understanding of the video content.

To highlight this distinction, we manually mark the most informative frames relative to the query using yellow stars. These frames are more temporally concentrated in LongVideoBench (around specific events) and more uniformly distributed across the timeline in Video-MME.

This difference helps explain why FOCUS achieves greater performance gains on LongVideoBench: our method assumes that frame-level relevance scores are i.i.d., a common setting in multi-armed bandit formulations. This assumption neglects temporal dependencies between video segments. Consequently, retrieval-based methods for keyframe selection typically require regularization (Tang et al., 2025; Yu et al., 2024) to promote diversity and ensure coverage.

If temporal dependencies between segments (arms) are taken into account, the problem setting shifts toward Lipschitz or metric bandits (Kleinberg et al., 2008; Bubeck et al., 2011), and contextual bandits (Chu et al., 2011; Agarwal et al., 2014). We leave such extensions to future work.

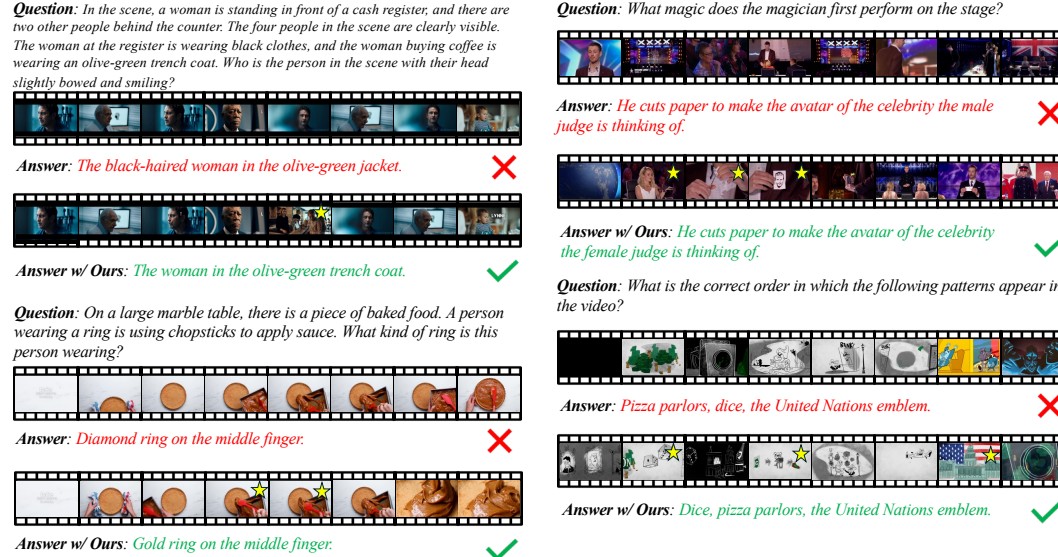

Figure 3: Comparison between uniformly sampled frames and those selected by FOCUS. The left column shows two examples from LongVideoBench; the right column shows two from Video-MME. Yellow stars indicate manually annotated frames that are most informative to the query, many of which are successfully captured by FOCUS.

## 3.3 COMPARISON WITH STATE-OF-THE-ART

| Method | LongVideoBench | | | | Video-MME | | | |
|---|---|---|---|---|---|---|---|---|
| | Short | Medium | Long | Overall | Short | Medium | Long | Overall |
| Uniform | 67.5 | 57.4 | 51.8 | 58.9 | 76.4 | 62.6 | 54.3 | 64.4 |
| Top-$K$ | **72.3** | 58.0 | 60.5 | 62.3 | 75.4 | 60.4 | 53.0 | 62.9 |
| AKS | **72.3** | **59.2** | 56.1 | 62.1 | 76.3 | 62.8 | 54.7 | 64.6 |
| **FOCUS (ours)** | **72.3** | 59.0 | **63.7** | **63.5** | **76.5** | **63.5** | **56.1** | **65.4** |

Table 2: Comparison between our method and state-of-the-art keyframe selection baselines under matched keyframe count. Results are reported by video length buckets: Short, Medium, and Long. For Video-MME, we adopt its original categorization: *Short* (<2 min), *Medium* (4-15 min), and *Long* (30-60 min). For LongVideoBench, we define *Short* as videos shorter than 3 minutes, *Medium* as 3-20 minutes, and *Long* as over 20 minutes to ensure a balanced distribution.

To further validate the effectiveness of FOCUS, we compare it against state-of-the-art training-free keyframe selection methods on both LongVideoBench and Video-MME. Specifically, we consider recent approaches based on vision-language similarity:

- **Top-$K$**: Computes relevance scores between each frame and the query, then selects the top-$K$ scoring frames. Due to computational constraints, we apply a pre-filtering step by downsampling videos to 1 frame per second.

- **AKS** (Tang et al., 2025): A recent method that adaptively balances frame relevance and temporal coverage. It is considered the current state-of-the-art and also incorporates pre-filtering via downsampling to 1 frame per second (Tang et al., 2025).

We also compare against Q-Frame (Zhang et al., 2025b), another recent training-free method that uses multi-resolution adaptation. Detailed comparisons across multiple MLLMs are provided in Section E, where FOCUS consistently outperforms both AKS and Q-Frame.

| Method | Filtering-free | Frames Seen (%) | GPU hours |
|---|---|---|---|
| AKS *w/o* pre-filtering | ✗ | 100 | 255 |
| AKS *w/* pre-filtering | ✗ | 3.7 | 9.3 |
| **FOCUS (Ours)** | ✔ | 1.6 | 5.5 |

Table 3: Efficiency comparison of keyframe selection methods on LongVideoBench. "Pre-filtering" refers to downsampling videos to 1 fps prior to selection. Note that the official AKS pipeline includes this pre-filtering step by default. "Frames Seen (%)" counts the proportion of frame-level BLIP forward passes relative to scoring all frames; GPU hours are measured on a single H100 (80GB).

**Fair comparison protocol.** We ensure a fair comparison by: (i) evaluating all methods using LLaVA-Video-7B, the best-performing MLLM in our setup; (ii) fixing the number of selected keyframes to $k = 64$; (iii) using the same vision-language encoder (e.g., BLIP) for frame scoring whenever possible. Results are summarized in Table 2.

**Consistency across different lengths.** FOCUS achieves consistent performance gains across all video length categories, with particularly strong improvements on long videos. On LongVideoBench, FOCUS outperforms uniform sampling by 11.9% and Top-$K$ by 7.6% on videos longer than 20 minutes. On Video-MME, the respective improvements are 1.8% and 1.4%.

We also observe that on short videos, all keyframe selection methods perform similarly and consistently outperform uniform sampling. We attribute this to a possible saturation in the reasoning capabilities of the underlying MLLM (LLaVA-Video-7B), where input selection plays a less critical role.

**Efficiency comparison.** We report the efficiency of each method in Table 3, measuring both the number of frames "seen" (i.e., scored by a vision-language model) and the total GPU hours required to perform keyframe selection. All GPU hours are measured using a single NVIDIA H100 (80GB) GPU on the LongVideoBench dataset.

As shown, AKS without pre-filtering is computationally infeasible in practice, as it requires scoring all video frames—amounting to over 255 GPU hours by the optimistic estimation. With pre-filtering, the cost drops significantly to 9.3 GPU hours. In contrast, FOCUS is the most efficient: it requires only 1.6% of the BLIP forward passes and just 5.5 GPU hours, while simultaneously achieving the best overall performance.

**Ablation Studies.** We conduct comprehensive ablation studies to validate key design choices, including the two-stage exploration-exploitation procedure, Bernstein confidence radius, clip length, and vision-language encoder selection. Results and detailed analysis are provided in Section G.

### 3.4 EFFICIENCY-ACCURACY TRADE-OFF

FOCUS exposes a natural trade-off between accuracy and computational cost through a single hyperparameter $\alpha$, which controls the fraction of arms selected for fine-grained exploration. We report accuracy and efficiency under different $\alpha$ settings in Table 4.

| | Accuracy (%) | Frames Seen (%) | GPU hours |
|---|---|---|---|
| $\alpha = 0.1$ | 62.9 | 1.1 | 3.5 |
| $\alpha = 0.25$ | 63.5 | 1.6 | 5.5 |
| $\alpha = 0.5$ | 63.6 | 2.5 | 9.2 |

Table 4: Effect of $\alpha$ on the performance and efficiency of FOCUS. "Frames Seen (%)" counts the proportion of frame-level BLIP forward passes relative to scoring all frames; GPU hours are measured on a single H100 (80GB).

We observe that choice of $\alpha$ has a significant impact on the efficiency while remain stable on the performance. With $\alpha = 0.1$, FOCUS evaluates 1.1% of frames and finishes in 3.5 GPU hours. At

$\alpha = 0.25$, the fraction rises to 1.6% with a cost of 5.5 GPU hours, yielding 63.5% accuracy. Setting $\alpha = 0.5$ achieves the highest accuracy (63.6%) but requires evaluating 2.5% of frames and 9.2 GPU hours—only a negligible gain over $\alpha = 0.25$ for a substantially higher cost, indicating diminishing returns from exploring more arms.

## 4  CONCLUSION

We addressed the core bottleneck of long-video understanding in MLLMs—the explosion of visual tokens—by introducing FOCUS, a training-free, plug-and-play keyframe selection method that allocates computation under a strict budget. FOCUS first partitions the video into temporal clips, treats each as an arm in a bandit problem, and then identifies query-relevant regions via a combinatorial pure-exploration strategy using empirical means and Bernstein confidence bounds. To improve efficiency, we reduce the iterative bandit process to a coarse-to-fine two-stage procedure that preserves optimism while enabling parallel inference.

Experiments on two challenging long-video QA benchmarks demonstrate that FOCUS consistently improves accuracy across four MLLMs while processing fewer than 2% of video frames. Our results show that lightweight, training-free keyframe selection—when guided by statistical principles—can significantly enhance the scalability and practicality of MLLMs for long-video understanding.

## 5  REPRODUCIBILITY STATEMENT

We provide a comprehensive theoretical analysis of our method in Appendix B and Appendix C. All models and datasets used in our study are publicly accessible.

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

# A APPENDIX

## A.1 RELATED WORK

### A.1.1 MULTIMODAL LARGE LANGUAGE MODELS (MLLMs) FOR VIDEO UNDERSTANDING

Recent MLLMs extend large language models with visual encoders, encoding images or frames into visual tokens that are fused with text to support open-ended video understanding. Most follow an encode-project-fuse pipeline with instruction tuning, as exemplified by the LLaVA family, Video-LLaVA/Video-LLaMA/Video-ChatGPT, and LLaMA-Vid/VideoChat (Liu et al., 2023; Lin et al., 2023; Zhang et al., 2023; Maaz et al., 2024; Li et al., 2024c;b). Progress has largely come from scaling data/backbones and strengthening cross-modal alignment (MiniCPM-V, InternVL/InternVL2, Qwen2-VL; data-centric and modality-binding advances via ShareGPT4Video and LanguageBind) (Yao et al., 2024; Chen et al., 2024e;d;c; Wang et al., 2024a; Chen et al., 2024a; Zhu et al., 2024), together with architectural refinements that unify multi-granularity visual inputs and tighten temporal adapters, and that improve projector efficiency or curricula (LLaVA-OneVision, LLaVA-NeXT/LLaVA-NeXT-Video, Aria, PLLaVA, Kangaroo) (Li et al., 2025; Liu et al., 2024a; Zhang et al., 2024c; Li et al., 2024a; Xu et al., 2024; Liu et al., 2024b). Finally, several models explicitly target extended context and hierarchical summarization for long-form understanding (LongVILA, LongVA, LongVLM, LongVU) (Chen et al., 2024b; Zhang et al., 2024b; Weng et al., 2024; Shen et al., 2024).

However, this tokenization-first paradigm encounters *token explosion* on long videos, where dense sampling yields prohibitive sequences. Recent efforts reduce the budget by compressing or restructuring tokens: MovieChat (Song et al., 2024) compacts frames into sparse memory, Video-XL-2 (Qin et al., 2025) synthesizes condensed tokens, and VideoStreaming (Qian et al., 2024) processes streams incrementally to cap tokens. Planning/tool-augmented agents (e.g., VideoAgent (Wang et al., 2024b)) curb perception via selective analysis, while hierarchical controllers (VideoTree (Wang et al., 2025)) and scaling recipes (VideoLLaMA 3 (Zhang et al., 2025a)) aid long-horizon reasoning. Beyond compression, ViLAMP (Cheng et al., 2025) uses mixed-precision tokenization to emphasize differential frames/patches and allocate capacity adaptively; long-context instruction-tuning such as Long-VITA (Shen et al., 2025) complements these strategies for long videos.

### A.1.2 VISION-LANGUAGE PRETRAINED MODELS

Cross-modal vision-language pretraining spans two-stream fusion, single-stream fusion, dual-encoder contrastive learning, and encoder-decoder hybrids. Two-stream models such as ViLBERT (Lu et al., 2019) and LXMERT (Tan & Bansal, 2019) encode vision and text separately and fuse via cross-attention, while single-stream counterparts—VisualBERT (Li et al., 2019), VL-BERT (Su et al., 2020), UNITER (Chen et al., 2020)—concatenate region features with text in a unified Transformer using MLM and alignment losses. Large-scale dual encoders like CLIP (Radford et al., 2021) and ALIGN (Jia et al., 2021) learn contrastive embeddings for zero-shot transfer, with FILIP (Yao et al., 2022) improving fine-grained patch-token alignment. Hybrid objectives combine contrastive and generative training (Li et al., 2021; Yu et al., 2022; Wang et al., 2022a; Chen et al., 2023) unify captioning and VQA. The BLIP family integrates vision encoders with language modeling—BLIP (Li et al., 2022) and BLIP-2 (Li et al., 2023) (via a lightweight Q-Former)—while Flamingo (Alayrac et al., 2022) and PaLM-E (Driess et al., 2023) inject visual inputs into large LMs for few-shot multimodal reasoning.

Extending to video, early pretraining models learned joint spatio-temporal-language representations with lightweight fusion and sparse sampling. VideoBERT (Sun et al., 2019) pairs frame sequences with transcripts in a BERT-style objective for retrieval and script generation, while HERO (Li et al., 2020) and ClipBERT (Lei et al., 2021) improve efficiency via hierarchical encoding and key-frame sampling for video-text retrieval and QA. Building directly on large image-text models, Clip4Clip (Luo et al., 2022) reuses CLIP encoders and matches videos to text via contrastive similarity, and FrozenBiLM (Yang et al., 2022) freezes a bi-directional LM while aligning a video encoder for zero-shot VQA.

### A.1.3 KEYFRAME SELECTION

In video representation learning, keyframe selection spans two major paradigms.

**Training-free keyframe selection.** Recent *training-free* methods leverage pretrained vision-language models and lightweight heuristics to pick informative, query-relevant frames. Adaptive Keyframe Sampling (AKS) maximizes prompt-frame similarity while enforcing temporal coverage via a split-and-judge policy (Tang et al., 2025); Q-Frame ranks frames by query-conditioned importance and preserves a few at higher resolution for detail (Zhang et al., 2025b). Text-frame alignment with frozen models further enables plug-and-play selectors (KeyVideoLLM, BOLT) that boost Video-LLM performance without fine-tuning (Liang et al., 2024; Liu et al., 2025). To avoid redundancy and preserve structure under a token budget, Logic-in-Frames performs dynamic, logic-verified search (Guo et al., 2025), while VideoTree builds a hierarchical, query-adaptive frame pyramid that expands salient scenes (Wang et al., 2025).

**Instruction-aligned and learned selectors.** Instruction-guided approaches train selectors with LLM/MLLM feedback: Frame-Voyager learns to query frame combinations by ranking sets with a pretrained Video-LLM (Yu et al., 2024), and Hu et al. (2025) supervise a lightweight selector using MLLM-derived single-frame relevance and multi-frame complementarity. Classical summarization remains relevant: supervised LSTM-based models (vsLSTM, dppLSTM; hierarchical RNNs) learn importance/diversity from human summaries (Zhang et al., 2016; 2018; Zhao et al., 2017), while unsupervised RL/adversarial methods (DR-DSN, SUM-GAN) optimize diversity-representativeness or realism without labels (Zhou et al., 2018; Mahasseni et al., 2017); however, these are typically task-agnostic and may miss frames critical for query-driven VQA.

### A.1.4 MULTI-ARMED BANDITS AND BATCHED EXPLORATION

Multi-armed bandits (MAB) encompass both regret minimization and pure exploration. Regret-oriented methods such as UCB variants and Thompson Sampling establish logarithmic-regret foundations for sequential decision-making (Auer et al., 2002; Lai & Robbins, 1985; Agrawal & Goyal, 2012). Pure exploration instead targets high-confidence identification with minimal samples, formalized as best-arm (and top-$k$) identification (Even-Dar et al., 2006; Bubeck et al., 2009; Kalyanakrishnan & Stone, 2010; Cao et al., 2015). Early elimination schemes (Successive/Median Elimination) provide PAC guarantees (Even-Dar et al., 2006; 2002), while confidence-bound and racing families—LUCB, UCB-E, and near-optimal lil'UCB—sharpen sample complexity and approach known lower bounds (Kalyanakrishnan et al., 2012; Audibert & Bubeck, 2010; Karnin et al., 2013; Jamieson et al., 2014; Kaufmann et al., 2016). Beyond single arms, combinatorial pure exploration (CPE) seeks an optimal subset under structural constraints, combining bandit confidence bounds with combinatorial oracles to search exponentially large spaces efficiently (Chen et al., 2016; Lattimore & Szepesvári, 2020).

Fully sequential adaptivity can be impractical when decisions must be made in few rounds or in parallel. Batched (parallel) bandits address this by operating over a small number of adaptivity rounds, yet retain near-sequential sample efficiency for pure exploration in theory and practice (Perchet et al., 2016; Jun et al., 2016; Gao et al., 2019). Batch-elimination/LUCB-style procedures match sequential complexity up to constants with only a handful of updates (Jun et al., 2016), and lower-bound trade-offs between batches and samples are well understood with matching algorithms (Perchet et al., 2016; Kaufmann et al., 2016; Tuynman & Degenne, 2025). Recent designs such as Tri-BBAI attain asymptotically optimal fixed-confidence BAI with just three batches, underscoring the feasibility of resource-constrained exploration (Jin et al., 2024).

## B  BERNSTEIN CONFIDENCE RADIUS

**Theorem B.1.** *Let $N_a(n)$ be the number of pulls for arm $a$ at round $n$ and $n = \sum_{a \in \mathcal{A}} N_a(n)$ is the total number of pulls. Let $\hat{\mu}_a(n)$ be the empirical mean of arm $a$ at round $n$ and $\hat{\sigma}_a^2(n)$ be the empirical variance of arm $a$ at round $n$. We define the empirical Bernstein Confidence Radius $\beta_a(n)$ as*

$$\beta_a(n) = \sqrt{\frac{2\,\hat{\sigma}_a^2 \ln n}{\max(1, N_a(n))}} + \frac{3 \ln n}{\max(1, N_a(n))}.$$

*Then we have the following bound holds with probability at least $1 - \frac{6}{n}$:*

$$|\hat{\mu}_a(n) - \mu_a| \le \beta_a(n)$$

*Proof.* Under the setting of frame-query relevance setting, the reward $r_t$ and latent frame reward $y_t$ is naturally bounded in $[0, 1]$. Therefore, according to Bernstein inequality, for any $\delta \in (0, 1)$, we have

$$\mathcal{P}\left[\mu_a \leq \hat{\mu}_a(n) + \sqrt{\frac{2\hat{\sigma}_a^2 \ln \frac{3}{\delta}}{N_a(n)}} + \frac{3 \ln \frac{3}{\delta}}{N_a(n)}\right] \geq 1 - \delta.$$

And symmetrically, we have

$$\mathcal{P}\left[\mu_a \geq \hat{\mu}_a(n) - \sqrt{\frac{2\hat{\sigma}_a^2 \ln \frac{3}{\delta}}{N_a(n)}} - \frac{3 \ln \frac{3}{\delta}}{N_a(n)}\right] \geq 1 - \delta.$$

Therefore, we have

$$\mathcal{P}\left[|\hat{\mu}_a(n) - \mu_a| \leq \sqrt{\frac{2\hat{\sigma}_a^2 \ln \frac{3}{\delta}}{N_a(n)}} + \frac{3 \ln \frac{3}{\delta}}{N_a(n)}\right] \geq 1 - 2\delta.$$

Choose $\delta = \frac{3}{n}$, then we have

$$|\mu_a - \hat{\mu}_a(n)| \leq \sqrt{\frac{2\hat{\sigma}_a^2 \ln \frac{3}{\delta}}{N_a(n)}} + \frac{3 \ln \frac{3}{\delta}}{N_a(n)}.$$

holds with probability at least $1 - \frac{6}{n}$.

When $N_a(n) = 0$, the statement is trivially true. Thus, we have the following bound holds with probability at least $1 - \frac{6}{n}$:

$$|\mu_a - \hat{\mu}_a(n)| \leq \beta_a(n).$$

$\square$

## C  REGRET BOUND

**Arm-level Regret Bound**

**Theorem C.1.** *Algorithm 2 returns the oracle top-$s$ set $S^\star$ with probability at least $1 - \frac{6M}{n}$ when terminated.*

*Proof.* When Algorithm 2 terminates, the following condition holds:

$$\max_{a \notin \hat{S}} \hat{\mu}_n(a) + \beta_a(n) \leq \min_{a \in \hat{S}} \hat{\mu}_n(a) - \beta_a(n).$$

According to Theorem B.1, with probability at least $1 - \frac{6}{n}$, we have $|\mu_a - \hat{\mu}_a(n)| \leq \beta_a(n)$ for all arms $a$. Therefore, for any $a \notin \hat{S}$,

$$\mathcal{P}\left[a \in S^\star\right] \leq 1 - \frac{6}{n}.$$

Thus, the probability that there does not exist such an arm $a$ is at least $1 - \frac{6(M-m)}{n}$, where $m$ is size of the $\hat{S}$ set. And this completes the proof. $\square$

**Frame-level Regret Bound**  We define the frame-level regret as the difference between the optimal frame-level reward and the reward of the selected frames.

$$r_N^{\text{frame}} = \sum_{t \in \mathbb{K}^\star} y_t - \sum_{t \in \hat{\mathbb{K}}_n} y_t.$$

As long as we obtain the oracle top-$s$ set $S^\star$, the frame-level regret is also guaranteed to be small. As Frame-level sampling is actually finite so we can always find the top-$k$ frames with the highest rewards.

$$\mathbb{E}r_N^{\text{frame}} = \mathbb{E}\sum_{t \in \mathbb{K}^\star} y_t - \sum_{t \in \hat{\mathbb{K}}_n} y_t = \mathbb{E}\sum_{a \in S^\star} \sum_{t \in \mathbb{K}_a^\star} 2\epsilon_\psi = 0.$$

For tighter bound, we leave this to future work.

*Question: When a pie chart representing the Czech Ethnicity appears in the video, with blue occupying the largest portion, red being the second, and light green the least, which of the following sentences is displayed on the screen?*

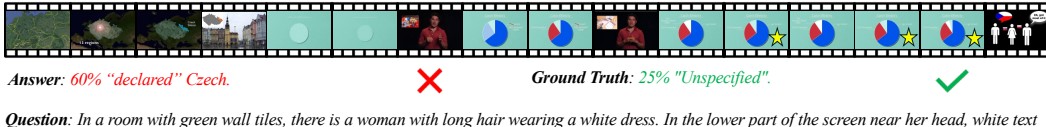

*Answer: 60% "declared" Czech.* ✗        *Ground Truth: 25% "Unspecified".* ✓

*Question: In a room with green wall tiles, there is a woman with long hair wearing a white dress. In the lower part of the screen near her head, white text appears that says 'someone started playing drums in the back.' What change happens to her when she appears in the restroom?*

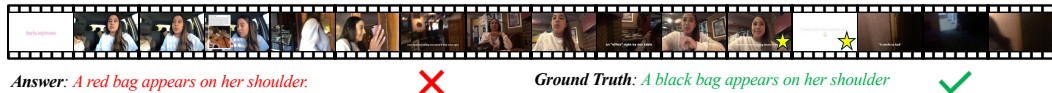

*Answer: A red bag appears on her shoulder.* ✗        *Ground Truth: A black bag appears on her shoulder* ✓

Figure 4: Two representative failure modes of LLaVA-Video-7B when using FOCUS to select keyframes. Yellow stars mark manually annotated frames that are most informative for the query. In the first case, FOCUS correctly selects these frames, but the MLLM still fails to answer due to its limited ability to reason over the relatively complex chart. In the second case, FOCUS fails to capture the critical frames during a compact, rapid scene transition: the relevant segment lasts only 1-2 seconds within a 10-minute video, making the keyframes difficult to identify even for human experts.

## D    VISUALIZATIONS OF FAILURE CASES

To provide a more comprehensive understanding of the proposed FOCUS, we analyze two typical failure patterns of LLaVA-Video-7B when using FOCUS to select keyframes in Figure 4, which most failure cases fall into.

In the first case, the query asks: "When a pie chart representing the Czech ethnicity appears in the video, with blue occupying the largest portion, red the second, and light green the least, which of the following sentences is displayed on the screen?" Across the entire video, this pie chart appears multiple times and is interleaved with other background content. Consequently, even though FOCUS correctly selects the most informative frames, the MLLM is confused by the subtle differences between multiple similar pie charts. This failure pattern is mainly attributable to the limited reasoning and perception capabilities of the MLLM itself, rather than to the keyframe selection method.

In the second case, the video is a 10-minute vlog with frequent scene transitions. The query asks: "In a room with green wall tiles, there is a woman with long hair wearing a white dress. In the lower part of the screen near her head, white text appears that says 'someone started playing drums in the back.' What change happens to her when she appears in the restroom?" The relevant segment lasts only 1–2 seconds within the 10-minute video, making the keyframes difficult to identify even for human experts. As shown in Figure 4, FOCUS successfully selects frames where the correct text appears, but still fails to capture the most critical frames. This pattern reveals that, in some intrinsically challenging cases, the adaptive sampling strategy of FOCUS may risk missing crucial information.

## E    COMPARISON WITH STATE-OF-THE-ART

Here we compare our proposed FOCUS against state-of-the-art training-free keyframe selection methods on both LongVideoBench and Video-MME. Specifically, we consider two recent approaches based on vision-language similarity:

- **AKS** (Tang et al., 2025): A plug-and-play adaptive keyframe sampling module that recursively balances query–frame relevance and temporal coverage under a fixed frame budget. By first downsampling the video to 1 frame per second, scoring each frame with a prompt–frame matching model, and then applying a judge-and-split procedure to allocate keyframe slots across segments, AKS maximizes informative coverage and serves as a strong state-of-the-art baseline for long-video QA.

- **Q-Frame** (Zhang et al., 2025b): A training-free, query-aware frame selection and multi-resolution adaptation framework that can be plugged in front of diverse Video-LLMs. It uses a text–image matching network (e.g., CLIP) to compute query–frame similarity scores, samples a compact set

| Model | #Frame | LLM | LongVideoBench | Video-MME |
|---|---|---|---|---|
| Qwen2-VL-7B | 32 | 7B | 55.6 | 57.4 |
| Qwen2-VL-7B *w/* AKS | 32 | 7B | 57.8 | **59.7** |
| Qwen2-VL-7B *w/* Q-Frame | 32 | 7B | 57.4 | 56.5 |
| Qwen2-VL-7B *w/* Ours | 32 | 7B | **62.3** ↑ 6.7 | **59.7** ↑ 2.3 |
| LLaVA-OV-7B | 32 | 7B | 54.8 | 56.5 |
| LLaVA-OV-7B *w/* AKS | 32 | 7B | 57.4 | 57.7 |
| LLaVA-OV-7B *w/* Q-Frame | 32 | 7B | 54.8 | 56.8 |
| LLaVA-OV-7B *w/* Ours | 32 | 7B | **60.7** ↑ 5.9 | **58.3** ↑ 1.8 |
| LLaVA-Video-7B | 64 | 7B | 58.9 | 64.4 |
| LLaVA-Video-7B *w/* AKS | 64 | 7B | 62.1 | 64.6 |
| LLaVA-Video-7B *w/* Q-Frame | 64 | 7B | 59.9 | 64.5 |
| LLaVA-Video-7B *w/* Ours | 64 | 7B | **63.5** ↑ 4.6 | **65.4** ↑ 1.0 |

Table 5: Video question-answering accuracy (%) of different MLLMs on LongVideoBench and Video-MME. We compare FOCUS with AKS and Q-Frame on Qwen2-VL, LLaVA-OV, and LLaVA-Video. The suffix "*w/* Ours" denotes models using keyframes selected by FOCUS; likewise, "*w/* AKS" and "*w/* Q-Frame" indicate using keyframes from the corresponding baselines. **#Frame** is the number of frames fed into the MLLM, and **LLM** denotes the language model size.

of highly relevant frames via stochastic selection, and assigns them heterogeneous resolutions so that crucial frames are preserved at high fidelity under a fixed token budget.

We report the results in Table 5. Across all three backbones and both benchmarks, FOCUS consistently outperforms both AKS and Q-Frame under the same frame budget. In particular, FOCUS improves the plain Qwen2-VL-7B, LLaVA-OV-7B, and LLaVA-Video-7B models by 4.6–6.7% on LongVideoBench and up to 2.3% on Video-MME, indicating that our keyframe selection strategy transfers robustly across different MLLMs.

For the two compared baselines, AKS consistently outperforms Q-Frame on both LongVideoBench and Video-MME whenever Q-Frame is evaluated. We attribute this to the more sophisticated and adaptive sampling scheme of AKS, which explicitly balances query–frame relevance and temporal coverage instead of relying solely on similarity scores.

By contrast, Q-Frame behaves more like a token-compression mechanism: it maps a fixed frame budget to a fixed number of visual tokens so that the MLLM can "see" more frames than it is originally designed for. However, the lack of an explicit temporal sampling or search design means that Q-Frame does not actively reason about where informative moments occur in long videos, which limits its performance in the long-form setting.

## F  EXPERIMENTS ON MORE BENCHMARKS

| Model | #Frame | LLM | MLVU | VSI-Bench |
|---|---|---|---|---|
| Qwen2-VL-7B | 32 | 7B | 59.7 | 36.5 |
| Qwen2-VL-7B *w/* AKS | 32 | 7B | 64.3 | 36.9 |
| Qwen2-VL-7B *w/* Ours | 32 | 7B | **67.0** ↑ 6.7 | **39.0** ↑ 2.5 |
| LLaVA-Video-7B | 64 | 7B | 68.2 | 41.7 |
| LLaVA-Video-7B *w/* AKS | 64 | 7B | 71.2 | 42.2 |
| LLaVA-Video-7B *w/* Ours | 64 | 7B | **72.7** ↑ 4.5 | **42.4** ↑ 0.7 |

Table 6: Video question-answering accuracy (%) of different MLLMs on MLVU and VSI-Bench. We compare FOCUS with AKS on Qwen2-VL and LLaVA-Video. The suffix "*w/* Ours" denotes models using keyframes selected by FOCUS; likewise, "*w/* AKS" indicates using keyframes from the corresponding baselines. **#Frame** is the number of frames fed into the MLLM, and **LLM** denotes the language model size.

To further investigate the generalization ability of FOCUS beyond long-form QA benchmarks, we conduct experiments on two additional datasets:

- **MLVU** (Zhou et al., 2025): A comprehensive multi-task long-video understanding benchmark constructed from 1,730 long videos (3 minutes to 2 hours) spanning movies, surveillance, egocentric recordings, cartoons, and game videos. It defines nine evaluation tasks that jointly probe both global and local reasoning abilities of MLLMs, and reveals substantial performance degradation as video length grows.
- **VSI-Bench** (Yang et al., 2025): A video-based visual–spatial intelligence benchmark built from 288 egocentric indoor videos (ScanNet, ScanNet++, ARKitScenes) with over 5,000 question–answer pairs. It focuses on 3D spatial understanding and memory from first-person streams, evaluating MLLMs on tasks such as spatial layout reasoning, navigation, and distance estimation.

We summarize the results in Table 6. On MLVU, our method improves Qwen2-VL-7B from 59.7% to 67.0% (+7.3%) and LLaVA-Video-7B from 68.2% to 72.7% (+4.5%), while also outperforming AKS by +2.7% and +1.5% points, respectively. On VSI-Bench, which emphasizes fine-grained spatial reasoning over relatively short egocentric clips, our method still yields consistent gains: for Qwen2-VL-7B, accuracy increases from 36.5% to 39.0% (+2.5%), and for LLaVA-Video-7B from 41.7% to 42.4% (+0.7%), respectively. These results indicate that our temporal search mechanism generalizes well across different backbones and tasks, with particularly pronounced benefits on long and heterogeneous videos.

At the same time, the improvements on VSI-Bench are understandably smaller than on long-video benchmarks. When videos are short and informative content is more uniformly distributed, uniform sampling already captures many salient frames, leaving less headroom for sophisticated temporal search. We explicitly regard this as a limitation and a promising direction for future work on spatially-aware frame selection in low-redundancy settings.

## G    ABLATION STUDIES

### G.1    TWO-STAGE EXPLORATION-EXPLOITATION

One of the core designs of FOCUS is the two-stage exploration-exploitation procedure. To better understand the contribution of each stage, we introduce two variants of FOCUS:

- **FOCUS-C**: This variant only performs the coarse exploration stage to identify promising temporal arms. In the final keyframe selection step, it randomly samples frames from all frames within the selected arms without any further refinement.
- **FOCUS-F**: This variant only performs the fine-grained exploration stage by uniformly sampling frames over the whole video and interpolating the rewards via nearest-neighbor assignment. The final keyframes are then drawn directly from the resulting video-level sampling distribution, without the arm-level pre-selection.

|  | Uniform | FOCUS-C | FOCUS-F | FOCUS |
|---|---|---|---|---|
| Qwen2-VL | 55.6 | 61.7 | 61.5 | **62.3** |
| LLaVA-OV | 54.8 | 58.4 | 57.7 | **60.7** |
| LLaVA-Video | 58.9 | 62.3 | 62.5 | **63.5** |

Table 7: Ablation of the two-stage exploration-exploitation procedure on LongVideoBench. **Uniform** denotes naive uniform frame sampling. **FOCUS-C** uses only the coarse exploration stage to select promising temporal arms, and then randomly samples frames within them. **FOCUS-F** uses only the fine-grained exploration stage over the entire video. The full **FOCUS** combines both stages and consistently achieves the best performance across all MLLMs, indicating that coarse arm selection and fine-grained refinement are complementary.

We conduct experiments on LongVideoBench with Qwen2-VL-7B, LLaVA-OV-7B, and LLaVA-Video-7B, and summarize the ablation results in Table 7. Both FOCUS-C and FOCUS-F provide substantial improvements over uniform sampling across all three backbones, demonstrating that

coarse arm selection and fine-grained exploration are each effective on their own. The full two-stage variant further yields the best performance in all cases, achieving an additional gain of up to 2.3% over the single-stage variants, which confirms that coarse localization of promising regions and subsequent fine-grained exploitation are complementary rather than interchangeable.

## G.2 BERNSTEIN CONFIDENCE RADIUS

Compared with the classical UCB algorithm, the Bernstein confidence radius is more robust to high-variance rewards. To better understand its contribution, we introduce a variant of FOCUS that relies on the empirical mean without a variance-aware exploration bonus when selecting top-relevance frames:

- **FOCUS-M**: This variant uses the empirical mean reward to rank arms and select top-relevance frames, instead of the Bernstein confidence radius.

|  | Uniform | FOCUS-M | FOCUS |
|---|---|---|---|
| Qwen2-VL | 55.6 | 61.7 | **62.3** |
| LLaVA-OV | 54.8 | 58.1 | **60.7** |
| LLaVA-Video | 58.9 | 63.0 | **63.5** |

Table 8: Ablation of the Bernstein confidence radius on LongVideoBench. **Uniform** denotes naive uniform frame sampling. **FOCUS-M** uses the empirical mean to rank arms and select top-relevance frames. The full **FOCUS** leverages the Bernstein confidence radius to form variance-aware upper confidence bounds.

We summarize the results in Table 8. The empirical-mean variant (FOCUS-M) already yields large gains over uniform sampling across all three backbones, showing that even a simple bandit-style selection is beneficial. However, the full method with the Bernstein confidence radius consistently achieves the best performance, providing up to 2.6% improvement over uniform and up to 2.6% improvement over the base models. This confirms that a variance-aware confidence radius is more effective than the empirical mean alone for selecting top-relevance frames, as it encourages additional exploration of high-uncertainty clips, especially when a clip contains diverse or rapidly changing scenes.

## G.3 EFFECT OF CLIP LENGTH

In the formulation of FOCUS, each video is partitioned into fixed-length clips that serve as bandit arms. The clip length $l$ is a crucial hyper-parameter that controls the granularity of exploration and exploitation. To better understand its effect, we conduct experiments on LongVideoBench with LLaVA-Video-7B and summarize the results in Table 9.

|  | Uniform | 8s | 16s | 32s |
|---|---|---|---|---|
| ACC | 58.9 | 63.7 | 63.5 | 62.3 |
| GPU hours | – | 8.1 | 5.5 | 4.1 |

Table 9: Ablation of the clip length $l$ on LongVideoBench with LLaVA-Video-7B. **Uniform** denotes naive uniform frame sampling (thus no additional GPU hours for keyframe selection are reported). For FOCUS, we vary the clip length from 8s to 32s and report both QA accuracy and the GPU hours required for keyframe selection. Note that the GPU hours are measured on a single NVIDIA H100 (80GB) GPU.

As shown in Table 9, all clip-length settings of FOCUS significantly outperform uniform sampling (58.9% vs. 62.3–63.7%), indicating that our bandit-based selection is robust to the choice of $l$ over a reasonably wide range. Shorter clips (e.g., 8s) provide slightly better accuracy by enabling more fine-grained exploration, but they also incur higher computational cost, while longer clips (e.g., 32s) reduce GPU hours at the price of a modest performance drop. In practice, we find $l = 16$ seconds to offer a good trade-off between accuracy and efficiency.

### G.4 Effect of Vision-Language Encoder

Our method can be seamlessly integrated with different vision-language encoders to estimate frame-query relevance scores. In the main experiments, we adopt BLIP to align with our primary baseline AKS for a fair comparison, and also because prior work has shown BLIP to be a robust and effective choice for frame-level relevance estimation. To provide a more comprehensive evaluation, we further conduct experiments with three encoders: CLIP (Radford et al., 2021), SigLIP (Zhai et al., 2023), and BLIP (Li et al., 2022).

|     | Uniform | CLIP | SigLIP | BLIP |
|-----|---------|------|--------|------|
| ACC | 58.9    | 60.2 | 60.9   | 63.5 |

Table 10: Ablation of the vision-language encoder on LongVideoBench with LLaVA-Video-7B. **Uniform** denotes naive uniform frame sampling. For our method, we instantiate the frame-query scoring module with CLIP, SigLIP, and BLIP.

As summarized in Table 10, all three encoders yield clear improvements over uniform sampling, confirming that our bandit-based selection is compatible with different vision-language backbones. Among them, BLIP achieves the strongest performance, while CLIP and SigLIP still provide 1.3% and 2.0% gains, respectively. These results suggest that our framework is robust to the choice of encoder, but can further benefit from stronger frame-query relevance models, and that future advances in vision-language pretraining are likely to directly translate into better keyframe selection performance.

## H Limitations

In this work, we assume the frame-query relevance scores are i.i.d. and the temporal dependencies between frames are not considered. However, in practice, the frame-query relevance scores are dependent on the temporal dependencies between frames. As different parts may have strong correlations, this assumption may not hold. In this setting, we can use the Lipschitz/metric bandit problem (Kleinberg et al., 2008; Bubeck et al., 2011) or contextual bandit problem (Chu et al., 2011; Agarwal et al., 2014) to model the problem. We leave this as future work.

## I The Use of Large Language Models (LLMs)

We used GPT-5 and Claude 4 solely for proofreading and light copy-editing (typos, grammar, and minor phrasing). All technical content, scientific claims, mathematical proofs, algorithms, experiment design and execution, dataset handling, figures, and evaluations were authored and verified by the human authors. LLMs were not used to generate ideas, code, data, results, or reviews; they did not contribute content at the level of a co-author. All suggested edits were manually inspected and accepted or rejected by the authors.

