# OpenReview forum: "FOCUS: Efficient Keyframe Selection for Long Video Understanding"
_ICLR.cc/2026/Conference — ICLR 2026 Poster_

### Official Review · Reviewer_aTVY · 2025-10-22

**Soundness:** 2
**Presentation:** 2
**Contribution:** 2
**Rating:** 6
**Confidence:** 3

**Summary:**

The paper proposes FOCUS (Frame-Optimistic Confidence Upper-bound Selection), a training-free, model-agnostic keyframe selection method for long video understanding with multimodal LLMs (MLLMs). To address the prohibitive token cost of processing all frames, FOCUS formulates keyframe selection as a combinatorial pure-exploration (CPE) problem in a multi-armed bandit setting, where short temporal clips are treated as arms. It employs a two-stage exploration–exploitation strategy: first using optimistic confidence upper bounds (UCB) to identify promising clips, then selecting top-scoring frames within them. Evaluated on LongVideoBench and Video-MME, FOCUS processes <2% of frames yet achieves consistent gains over uniform sampling and SOTA retrieval-based methods, e.g., +11.9% accuracy on videos >20 minutes. The method is simple, theoretically grounded, and plug-and-play compatible with existing MLLMs

**Strengths:**

The paper introduces FOCUS, a training-free keyframe selection method that formulates the task as a combinatorial pure-exploration bandit problem. It partitions the video into clips (arms), uses optimistic confidence upper bounds to identify informative regions with minimal sampling, and then selects top frames within those clips. This two-stage exploration-exploitation strategy enables MLLMs to achieve strong performance on long-video QA benchmarks while processing fewer than 2% of frames. It significantly outperforms uniform sampling and existing retrieval-based methods, especially on videos over 20 minutes.

**Weaknesses:**

- The concept of ACF and the calculation of $r_t$ in Figure 1 require further explanation, which will help readers better understand the motivation of the paper.

- This pre-filtering process before keyframe selection undermines the goal of identifying the most informative keyframes from all frames. The findings are interesting, but it is not certain whether the two-stage ARM selection proposed in the article will fall into the same limitations.

- Lack of experimental results on MLVU, a commonly used long video understanding benchmark.

- Minor Weaknesses
  - Line 36: multimodal LLMs (MLLMs) -> multimodal large language models (MLLMs)
  - Line 107: multimodal LLMs -> MLLMs

[1] Zhou J, Shu Y, Zhao B, et al. Mlvu: A comprehensive benchmark for multi-task long video understanding[J]. arXiv e-prints, 2024: arXiv: 2406.04264.

**Questions:**

- FOCUS first partition the timeline into $M$ non-overlapping fixed-length clips. Does this destroy the spatiotemporal consistency of the video? Which makes it difficult to capture continuous segments with high information density?

- As discussed in the Limitation section, FOCUS assumes that each frame of the video is independent, which is completely contrary to the nature of the video. Does this limit FOCUS’s applicability to short videos?

- Table 1 lacks the experimental results of other frame selection methods, such as AKS and Q-Frame, based on the same baseline.

- The paper provides a visualization of FOCUS's superiority over uniform sampling in Figure 3. It is meaningful to include the negative aspects of FOCUS, which helps readers better understand its limitations.

---

> ### Author Response · Authors · 2025-11-21
> **Response to Reviewer aTVY**
>
> We are grateful for your careful reading and constructive feedback. Below we address each of your comments and questions in detail.
>
> > **W1**: The concept of ACF and the calculation of r in Figure 1 require further explanation.
>
> **Re: Thank you for highlighting this point** and we also realized that the original explanation of the ACF and the relevance scores $r_t$ in Figure 1 was too brief.
>
> In the revised manuscript (Section 1 and the caption of Figure 1), we now explicitly define $r_t$ as the cosine similarity between the query text embedding and each frame embedding produced by a frozen vision–language encoder. We then introduce the autocorrelation function (ACF) as $\rho(\delta) = \mathrm{corr}(r_t, r_{t+\delta})$ over time lags $\delta$. We also clarify that the dashed line in Figure 1 corresponds to $\rho(\delta) = 0.5$, which we use to read off the correlation half-life (about 5 seconds). This empirical half-life is what motivates our bandit-style keyframe selection under strong temporal locality.
>
>
> > **W2**: This pre-filtering process before keyframe selection undermines the goal of identifying the most informative keyframes from all frames. The findings are interesting, but it is not certain whether the two-stage arm selection proposed in the article will fall into the same limitations.
>
> **Re: We appreciate this sharp and important comment.** Let us clarify what we mean by “pre-filtering” in the paper and how FOCUS differs from it.
>
> In our usage, **pre-filtering** refers specifically to *query-agnostic uniform downsampling* applied *before* keyframe selection method (e.g., reducing an hour-long video to 1 fps and then running a keyframe method only on those sampled frames). Such uniform pre-filtering can indeed undermine the goal of keyframe selection, because potentially informative frames may be irrevocably discarded without looking at the query.
>
> From a broader perspective, you are right that any keyframe selection method can be viewed as a form of “pre-filtering before the downstream MLLM,” since all of them ultimately select a subset of frames to reduce inference cost. Our intention was **not** to argue against pre-filtering in this general sense, but to distinguish:
>
> - *Crude, query-agnostic uniform pre-filtering*, which **discards frames before seeing any relevance signal**;
> - Versus *query-aware, adaptive selection*, which decides which frames to keep based on estimated utility.
>
> FOCUS belongs to the second category. All frames remain in the candidate pool, grouped into short clips (arms) purely for computational convenience. The bandit algorithm then adaptively reallocates scoring budget across arms based on query-aware BLIP scores and confidence radii, rather than uniformly discarding frames upfront.
>
>
>
> > **W3**: Lack of experimental results on MLVU, a commonly used long video understanding benchmark.
>
> **Re: Thank you for this suggestion.** We have added MLVU to our experiments and report the results in Appendix F. For completeness, we reproduce them here:
>
> |             | Frames | Uni  | AKS  | Ours |
> | ----------- | ------ | ---- | ---- | ---- |
> | Qwen2-VL    | 32     | 59.7 | 64.3 | 67.0 |
> | LLaVA-OV    | 32     | 15.3 | 19.3 | 22.7 |
> | LLaVA-Video | 64     | 68.2 | 71.2 | 72.7 |
>
> On MLVU, our method improves Qwen2-VL-7B from 59.7% to 67.0% (+7.3 points) and LLaVA-Video-7B from 68.2% to 72.7% (+4.5 points), while also outperforming AKS on all three backbones. We further include results on VSI-Bench in Appendix F.
>
> Together, these additional benchmarks show that our temporal search mechanism generalizes well across different models and long-video tasks.
>
>
> > **W4**: Minor weaknesses.
>
> **Re: Thank you for pointing these out.** We have corrected the noted issues and carefully proofread the revised manuscript for similar minor inconsistencies.

---

> ### Author Response · Authors · 2025-11-21
> **Response to Reviewer aTVY**
>
> > **Q1**: FOCUS first partitions the timeline into \(M\) non-overlapping fixed-length clips. Does this destroy the spatiotemporal consistency of the video, making it difficult to capture continuous segments with high information density?
>
> **Re: We appreciate this constructive question.** We clarify both how clips are defined and how our method behaves on dense, high-information segments.
>
> **(a) Adaptive number of clips, guided by ACF.**
> In practice, we do not fix a global \($M$) across videos. Instead, we use a fixed clip length \($l$\), so \($M$\) naturally adapts to each video’s duration. The choice of \($l$\) is guided by the ACF decay in Figure 1 (empirical half-life ~5s); in experiments we find \($l=16$s\) seconds works well. This keeps clips short enough that locally coherent segments (including high-information intervals) typically fall within one or a few neighboring arms, preserving temporal locality rather than destroying it.
>
> **(b) Behavior on continuous high-information segments.**
> Our method naturally handles shot changes and scene transitions. The Bernstein confidence radius scales with the empirical variance $\hat\sigma_a^2$ of each arm. If a fixed-length clip happens to straddle a scene transition boundary, the within-arm variance increases, the confidence interval widens, and the algorithm is encouraged to explore that uncertain arm more in the fine stage. This behavior follows the “optimism in the face of uncertainty” principle and is precisely what variance-adaptive UCB variants are designed to exploit. This is also why we call our method “Frame-Optimistic Confidence Upper-bound Selection”.
>
> In summary, the fixed-length partitioning does not destroy spatiotemporal structure: clips are segmented according to the ACF-guided length to preserve local temporal coherence, adapt their count to each video’s duration, and act as units for efficient exploration, while dense high-information segments are still prioritized by our mean- and variance-aware bandit mechanism.
>
>
> > **Q2**: As discussed in the Limitation section, FOCUS assumes that each frame of the video is independent, which is completely contrary to the nature of the video. Does this limit FOCUS’s applicability to short videos?
>
> **Re: Thank you for this insightful question.** We clarify both the distributional assumption and the short-video behavior.
>
> **(a) On the “independence” assumption.**
> In Section 2.2, we do not assume that contiguous frames are temporally i.i.d., nor that the whole video is globally stationary. Our assumption is **arm-wise and finite-population**:
>
> - For an arm (a short clip), the latent frame utilities form a finite set.
> - Pulling arm is modeled as drawing a frame uniformly from this set, with utility $y_t \sim \nu_a.$
>
> Our analysis and algorithm only require boundedness and the variance of $\nu_a$, under which Bernstein-style confidence radii remain valid. The Limitation section is referring to long-range dependence across clips (arms), not to independence at the frame level. Extending the theory to explicitly model cross-arm dependence (e.g., via contextual or structured bandits) is a promising direction for future work. We have revised Section 2.2 and Appendix D to make this distinction clearer and avoid the impression that we assume fully independent frames.
>
> **(b) Applicability to short videos.**
> For short videos, we provide a length-wise analysis in Section 3.3, with detailed results in Table 2. Empirically, FOCUS:
>
> - Achieves larger gains on longer videos.
> - Still improves performance on short videos (< 3 min) by about 4.8% over uniform sampling.
>
> This behavior matches intuition: for short clips, uniform sampling already covers many informative frames, so the room for improvement is smaller, but our method remains beneficial rather than breaking down. We will highlight this observation more explicitly in the revised text.

---

> ### Author Response · Authors · 2025-11-21
> **Response to Reviewer aTVY**
>
> > **Q3**: Table 1 lacks the experimental results of other frame selection methods, such as AKS and Q-Frame, based on the same baseline.
>
> **Re: Thank you for this suggestion.** In the revised manuscript, we add a new comparison table (now Table 5) including AKS and Q-Frame under the same backbones (Qwen2-VL-7B, LLaVA-OV-7B, and LLaVA-Video-7B) on both LongVideoBench and Video-MME. For convenience, we reproduce the results here:
>
> |                        | Frames | LLM  | LongVideoBench | VideoMME |
> | ---------------------- | ------ | ---- | -------------- | -------- |
> | Qwen2-VL               | 32     | 7B   | 55.6           | 57.4     |
> | Qwen2-VL w/ AKS        | 32     | 7B   | 57.8           | **59.7** |
> | Qwen2-VL w/ Q-Frame    | 32     | 7B   | 57.4           | 56.5     |
> | Qwen2-VL w/ Ours       | 32     | 7B   | **62.3**       | **59.7** |
> | LLaVA-OV               | 32     | 7B   | 54.8           | 56.5     |
> | LLaVA-OV w/ AKS        | 32     | 7B   | 57.4           | 57.7     |
> | LLaVA-OV w/ Q-Frame    | 32     | 7B   | 54.8           | 56.8     |
> | LLaVA-OV w/ Ours       | 32     | 7B   | **60.7**       | **58.3** |
> | LLaVA-Video            | 64     | 7B   | 58.9           | 64.4     |
> | LLaVA-Video w/ AKS     | 64     | 7B   | 62.1           | 64.6     |
> | LLaVA-Video w/ Q-Frame | 64     | 7B   | 59.9           | 64.5     |
> | LLaVA-Video w/ Ours    | 64     | 7B   | **63.5**       | **65.4** |
>
> Across all three backbones and both benchmarks, our method consistently outperforms AKS and Q-Frame under the same frame budget. In particular, it improves the plain Qwen2-VL-7B, LLaVA-OV-7B, and LLaVA-Video-7B models by about 4.6–6.7% on LongVideoBench and up to 2.3% on Video-MME, indicating that our keyframe selection strategy transfers robustly across different MLLMs.
>
>
> > **Q4**: The paper provides a visualization of FOCUS's superiority over uniform sampling in Figure 3. It is meaningful to include the negative aspects of FOCUS, which helps readers better understand its limitations.
>
> **Re: Thank you for this constructive suggestion.** In the revision, we explicitly visualize and discuss the limitations of FOCUS by adding a new section “Visualizations of Failure Cases” and a new Figure 5.
>
> We present two representative failure patterns:
> 1. **MLLM-limited cases (not a failure of FOCUS itself):** In a chart-understanding example, FOCUS correctly selects frames showing the relevant pie charts, but LLaVA-Video-7B still fails because it cannot reliably distinguish subtle differences across several similar charts. This indicates that some errors are due to the MLLM’s own reasoning/visual limitations rather than the keyframe selector.
>
> 2. **Intrinsically challenging cases for adaptive sampling:** In a 10-minute vlog with frequent scene transitions, the supporting evidence appears only for 1–2 seconds. Even though FOCUS captures frames with the correct on-screen text, it can still miss the exact critical moment. This illustrates that, in some intrinsically challenging scenarios with extremely brief events, the adaptive sampling strategy of FOCUS may risk missing crucial information.
>
> These visualizations clarify when FOCUS is likely to fail, giving a more balanced view of its strengths and limitations.

---

> ### Author Response · Authors · 2025-12-02
> **Summary for Reviewer aTVY**
>
> We thank Reviewer aTVY for their helpful comments. To help the AC and PCs quickly see how we addressed these concerns, we briefly summarize our responses and revisions below:
>
> 1. **ACF definition and motivation (W1).** We clarified the definition of the relevance scores $r_t$ (cosine similarity between query and frame embeddings) and the ACF $\rho(\delta) = \mathrm{corr}(r_t, r_{t+\delta})$ in Section 1 and the caption of Figure 1, and explained how the empirical half-life (≈5s) motivates our bandit-style keyframe selection.
>
> 2. **Pre-filtering vs. FOCUS’s adaptive selection (W2).** We clarified that “pre-filtering” refers to crude, query-agnostic uniform downsampling before any relevance scoring, and distinguished this from FOCUS’s query-aware, bandit-based selection: all frames remain candidates (grouped into short clips), and the algorithm adaptively allocates scoring budget instead of discarding frames uniformly upfront.
>
> 3. **Additional benchmarks and baselines (W3, Q3).** We added experiments on MLVU (and report VSI-Bench in the appendix) and included AKS and Q-Frame in the main comparison table under the same backbones on LongVideoBench and Video-MME, showing that FOCUS consistently improves over uniform sampling, AKS, and Q-Frame.
>
> 4. **Assumptions, video length, and clip partitioning (Q1, Q2).** We clarified that our assumption is arm-wise finite-population (the earlier “independence” remark referred to long-range dependence across clips), summarized our length-wise analysis (larger gains on longer videos but still noticeable improvements on short ones), and explained how a fixed clip length $\ell$ (guided by the ACF decay) balances temporal coherence with coverage as $M$ adapts to video duration.
>
> 5. **Failure cases and minor issues (Q4, W4).** We added a “Visualizations of Failure Cases” section with representative examples (MLLM-limited and brief-event failures) and corrected the typos and minor issues noted by the reviewer, including the MLVU citation.
>
> We believe these clarifications, additional experiments, and visualizations directly address Reviewer aTVY’s concerns about motivation, evaluation, and limitations.
>
> We sincerely thank the AC and PCs for their time and careful consideration of our submission.
>
> Authors of submission-18064

---

### Official Review · Reviewer_YVCf · 2025-10-25

**Soundness:** 3
**Presentation:** 3
**Contribution:** 3
**Rating:** 4
**Confidence:** 4

**Summary:**

This paper introduces FOCUS, a training-free and model-agnostic keyframe selection method for long-video understanding with multimodal large language models (MLLMs). FOCUS frames keyframe selection as a Combinatorial Pure Exploration (CPE) problem under a multi-armed bandit (MAB) formulation, treating short temporal clips as “arms.” By applying a Bernstein-style upper confidence bound (UCB-V) and a two-stage coarse-to-fine exploration scheme, the method efficiently identifies query-relevant temporal regions before selecting keyframes within each. Experiments on LongVideoBench and Video-MME, across four MLLMs (GPT-4o, Qwen2-VL, LLaVA-OV, and LLaVA-Video), show that FOCUS achieves 3–6% accuracy improvements over uniform sampling and comparable or slightly higher performance than recent training-free methods (AKS, Top-K), while processing only ~2% of frames.

**Strengths:**

- **Conceptually elegant and efficient.** The formulation connects keyframe selection with variance-adaptive exploration in MABs, offering a lightweight theoretical lens for efficient inference. The two-stage batched procedure is practical and easily parallelizable, reducing GPU cost by 40–60% compared to AKS.

- **Training-free and modular.** The pipeline is plug-and-play, requires no fine-tuning, and integrates smoothly into existing LVLM inference workflows.

- **Empirically consistent.** Evaluations span multiple datasets, video lengths, and MLLM architectures, with consistent accuracy and efficiency trade-offs (Table 1–4).

- **Clear presentation.** The paper is well-written with informative figures and pseudocode; Algorithm 2 is easy to follow.

**Weaknesses:**

- **Incremental novelty.** The “combinatorial pure-exploration bandit” framing is conceptually sound but reuses standard UCB-V principles with minimal adaptation to video reasoning. Similar adaptive sampling ideas have appeared in AKS, Q-Frame, T* and Frame-Voyager.

- **Weak theoretical substance & Limited methodological depth.** The regret bound assumes i.i.d. frame rewards and bounded noise, which do not hold in temporally correlated videos. The theoretical claim does not extend to frame-level optimality, limiting its practical significance. Despite the theoretical motivation, the final algorithm reduces to a fixed two-stage heuristic controlled by α and lacks adaptive or learnable exploration scheduling.

- **Missing core ablations.** The paper does not isolate contributions from key components—coarse vs. fine stages, confidence bounds vs. mean-based selection—so the source of improvement remains unclear.

- **Incomplete experimental comparison.** Evaluation omits recent reasoning-driven or event-centric baselines (e.g., Q-Frame, Logic-in-Frames ...), providing only partial evidence of superiority.

- **Lack of robustness and qualitative analysis.** No systematic study of failure cases, sensitivity to α or clip length, or behavior on complex multi-event queries is provided, leaving generalization uncertain.

**Questions:**

- Include ablation studies disentangling the effect of each module (coarse/fine, α, confidence radius).
- Compare against stronger or more diverse baselines (Q-Frame, Frame-Voyager, Logic-in-Frames, VSLS, $T^{*}$, Nar-KFC, TimeSearch).
- Evaluate robustness on longer and more diverse benchmarks, such as MLVU and LVBench (general), Ego4D/ EgoSchema (egocentric reasoning), and OVO-Bench / HLV-1K (reasoning).
- Analyze failure cases and provide qualitative insights on when FOCUS fails (e.g., multi-event reasoning).
- Consider extending the method with lightweight learnable scoring or adaptive thresholding to move beyond static heuristics.

---

> ### Author Response · Authors · 2025-11-21
> **Response to Reviewer YVCf**
>
> The reviewer's thoughtful comments are appreciated, and we take this opportunity to respond to the concerns raised, offering additional insights and clarifications.
>
> > **W1**: Incremental novelty. The “combinatorial pure-exploration bandit” framing is conceptually sound but similar adaptive sampling ideas have appeared in AKS, Q-Frame, T* and Frame-Voyager.
>
> **Re: We appreciate this thoughtful comment** and clarify that although prior work explores various forms of “adaptivity”, they differ fundamentally from our formulation. To highlight these distinctions, we summarize the comparison below:
>
>
> | Method           | Training-free | Model-agnostic | No reliance on pre-filtering | No teacher/ offline labels | Adaptive temporal exploration | Theory-grounded |
> | ---------------- | ------------- | -------------- | ---------------------------- | -------------------------- | ----------------------------- | --------------- |
> | **FOCUS (ours)** | ✅             | ✅              | ✅                            | ✅                          | ✅                             | ✅               |
> | AKS              | ✅             | ✅              | ❌                            | ✅                          | ❌                             | ❌               |
> | Q-Frame          | ✅             | ✅              | ❌                            | ❌                          | ❌                             | ❌               |
> | T\*              | ✅             | ❌              | ❌                            | ❌                          | ✅                             | ❌               |
> | Frame-Voyager    | ❌             | ❌              | ❌                            | ❌                          | ❌                             | ❌               |
>
> AKS and Q-Frame rely on uniform pre-filtering and then exhaustive scoring or resolution allocation. T* performs object-centric search using an oracle MLLM on uniformly sampled frames, and Frame-Voyager trains a selector tied to a specific video-LLM. None of these methods perform **temporal exploration with uncertainty quantification**.
>
> By contrast, **FOCUS is explicitly a training-free, plug-and-play, efficient exploration-exploitation selector**:
> (i) it adaptively reallocates scoring budget using variance-aware confidence bounds,
> (ii) requires no teacher supervision or oracle MLLM, and
> (iii) provides arm-level theoretical guarantees under the finite-population assumption.
> We will clarify these distinctions more explicitly in the related-work section.
>
>
> > **W2**: Weak theoretical substance & limited methodological depth. The regret bound assumes i.i.d. frame rewards and bounded noise, which do not hold in temporally correlated videos. The theoretical claim does not extend to frame-level optimality, limiting its practical significance. Despite the theoretical motivation, the final algorithm reduces to a fixed two-stage heuristic controlled by $\alpha$ and lacks adaptive or learnable exploration scheduling.
>
> **Re: We appreciate the comment** and hereby clarify our theoretical assumptions and algorithmic choices.
>
> **(1) Distributional assumption.**
> We do not assume that contiguous frames are temporally i.i.d., nor that the full video is globally stationary. Our assumption is arm-wise and finite-population: for an arm $a$ (a short clip), the latent frame utilities form a finite set, and pulling $a$ reveals the utility of a uniformly drawn frame from that set, i.e., $y_t \sim \nu_a$. Our analysis relies only on boundedness and the variance of $\nu_a$, under which Bernstein confidence radii remain valid. We have updated Section 2.2 and Appendix H to make this explicit and avoid confusion.
>
>
> **(2) Bounded noise.**
> This assumption holds naturally in our setting, since all frame–query relevance scores are bounded within $[0,1]$. Consequently, the noise magnitude is automatically bounded by this interval.
>
> **(3) Frame-level optimality.**
> While our formal guarantee is stated at the **arm level**, it naturally extends to the **frame level**. As discussed in Section 2.3, each arm (~16s video clip) contains a finite set of frames. Once the optimal arms are identified, evaluating all frames within those arms trivially recovers the optimal frame-level keyframes. We omitted this straightforward extension for brevity but will add a short explanation in the revised manuscript.
>
> **(4) Two-stage design.**
> The two-stage structure is a computational design choice, not a theoretical restriction. Fully adaptive or learned schedules would introduce unpredictable per-video sampling cost and hinder parallelization. Our two-stage variant preserves the CPE motivation (coarse elimination + fine exploration) while ensuring stable and efficient batching on modern accelerators. We will clarify this trade-off between methodological flexibility and practical efficiency in the revised version.

---

> ### Author Response · Authors · 2025-11-21
> **Response to Reviewer YVCf**
>
> > **W3**: Missing core ablations. The paper does not isolate contributions from key components—coarse vs. fine stages, confidence bounds vs. mean-based selection—so the source of improvement remains unclear.
>
> **Re: We appreciate this valuable suggestion**. In the revised version, we have added the requested ablations in Appendix G. Below we summarize the key findings.
>
> **(1) Coarse vs. fine stages.**
> To disentangle the contributions of the two-stage procedure, we introduce two variants:
>
> - **FOCUS-C:** Only performs the coarse exploration stage. After identifying promising arms, frames are uniformly sampled within those arms without additional refinement.
> - **FOCUS-F:** Skips coarse exploration entirely. It uniformly samples frames across the whole video, assigns rewards via nearest-neighbor interpolation, and draws keyframes from this video-level distribution.
>
> Results on three backbones (Qwen2-VL-7B, LLaVA-OV-7B, LLaVA-Video-7B) are shown below:
>
> |             | Uniform | FOCUS-C | FOCUS-F | FOCUS |
> | ----------- | ------- | ------- | ------- | ----- |
> | Qwen2-VL    | 55.6    | 61.7    | 61.5    | 62.3  |
> | LLaVA-OV    | 54.8    | 58.4    | 57.7    | 60.7  |
> | LLaVA-Video | 58.9    | 62.3    | 62.5    | 63.5  |
>
> Both stages independently outperform uniform sampling, and the full method consistently yields the highest accuracy, confirming that coarse localization and fine exploration are complementary.
>
> **(2) Confidence radius vs. empirical-mean selection.**
> We further study the benefit of using a Bernstein-style confidence bound by introducing **FOCUS-M**, which ranks arms purely by empirical mean without a variance-aware bonus.
>
> |             | Uniform | FOCUS-M | FOCUS |
> | ----------- | ------- | ------- | ----- |
> | Qwen2-VL    | 55.6    | 61.7    | 62.3  |
> | LLaVA-OV    | 54.8    | 58.1    | 60.7  |
> | LLaVA-Video | 58.9    | 63.0    | 63.5  |
>
> FOCUS-M already improves over uniform sampling, showing that even simple bandit-style selection is effective. However, FOCUS with Bernstein confidence radii consistently achieves the best performance. This confirms the value of variance-aware exploration, especially in clips containing heterogeneous or rapidly changing content.
>
> We have integrated these ablations and their discussion into the revised manuscript.
>
>
>
>
>
> > **W4**: Incomplete experimental comparison. Evaluation omits recent reasoning-driven or event-centric baselines (e.g., Q-Frame, Logic-in-Frames ...), providing only partial evidence of superiority.
>
> **Re: We appreciate this suggestion** and agree that including stronger baselines is important for a fair assessment.
>
> We have now added **Q-Frame** as an additional comparison point. The table below reports the results on LongVideoBench, and additional results are provided in Appendix E.
>
> |             | Frames | Uniform | Qframe | AKS  | Ours |
> | ----------- | ------ | ------- | ------ | ---- | ---- |
> | Qwen2-VL    | 32     | 55.6    | 57.4   | 57.8 | 62.3 |
> | LLaVA-OV    | 32     | 54.8    | 54.8   | 57.4 | 60.7 |
> | LLaVA-Video | 64     | 58.9    | 59.9   | 62.1 | 63.5 |
>
> Across all three backbones, our method consistently outperforms both the uniform baseline and Q-Frame, as well as AKS, providing stronger evidence of its effectiveness.
>
> For **Logic-in-Frames**, the official code is not publicly available. We attempted a careful reimplementation based on the paper, but the performance we obtained was consistently worse than uniform sampling. Our impression is that the four manually defined temporal relations in their design are quite sensitive and require implementation details that are not fully specified. To avoid reporting potentially misleading numbers from an imperfect reimplementation, we chose not to include Logic-in-Frames in the main tables, and we will add a short note in the revised manuscript to clarify our efforts.

---

> ### Author Response · Authors · 2025-11-21
> **Response to Reviewer YVCf**
>
> > **W5**: Lack of robustness and qualitative analysis. No systematic study of failure cases, sensitivity to $\alpha$ or clip length, or behavior on complex multi-event queries is provided, leaving generalization uncertain.
>
> **Re: We appreciate this important point** and have expanded the robustness analysis along three axes in the revised version.
>
> **(1) Failure cases and complex multi-event queries.**
> We add a new section “Visualizations of Failure Cases” and a new Figure 5, where we analyze two representative failure patterns that explicitly target multi-event/complex-query scenarios:
> 1. **MLLM-limited cases (not a failure of FOCUS itself):** In a chart-understanding example, FOCUS correctly selects frames showing the relevant pie charts, but LLaVA-Video-7B still fails because it cannot reliably distinguish subtle differences across several similar charts. This indicates that some errors are due to the MLLM’s own reasoning/visual limitations rather than the keyframe selector.
>
> 2. **Intrinsically challenging cases for adaptive sampling:** In a 10-minute vlog with frequent scene transitions, the supporting evidence appears only for 1–2 seconds. Even though FOCUS captures frames with the correct on-screen text, it can still miss the exact critical moment. This illustrates that, in some intrinsically challenging scenarios with extremely brief events, the adaptive sampling strategy of FOCUS may risk missing crucial information.
>
> These cases clarify when FOCUS tends to fail (complex multi-event narratives and extremely brief events in very long videos) and provide qualitative insight into its limitations.
>
> **(2) Sensitivity to $\alpha$ (efficiency–accuracy trade-off).**
> As discussed in Section 3.4, we have provided a systematic study of $\alpha$:
>
> | $\alpha$ | Accuracy (%) | Frames Seen (%) | GPU hours |
> | -------- | ------------ | --------------- | --------- |
> | 0.10     | 62.9         | 1.1             | 3.5       |
> | 0.25     | 63.5         | 1.6             | 5.5       |
> | 0.50     | 63.6         | 2.5             | 9.2       |
>
> Accuracy remains relatively stable across $\alpha$, while compute grows steadily. In practice, $\alpha= 0.25$ offers a good balance, as larger values bring only marginal accuracy gains at significantly higher cost. We highlight this trend more explicitly in the revised text.
>
> **(3) Sensitivity to clip length.**
> We further add an ablation on the clip length $l$ on LongVideoBench with LLaVA-Video-7B:
>
> |           | Uniform | 8s   | 16s  | 32s  |
> | --------- | ------- | ---- | ---- | ---- |
> | ACC       | 58.9    | 63.7 | 63.5 | 62.3 |
> | GPU hours | —       | 8.1  | 5.5  | 4.1  |
>
> All clip-length settings substantially outperform uniform sampling (58.9% vs. 62.3–63.7%), indicating that our bandit-based selection is reasonably robust to l in this range. Shorter clips (8s) offer slightly higher accuracy but higher cost; longer clips (32s) save GPU hours with a modest performance drop. In practice, we find $l$ = 16 seconds to be a good accuracy–efficiency compromise. We now discuss these observations explicitly in the paper.
>
> Together, these additions provide a more systematic view of robustness (failure modes, $\alpha$, and $l$) and clarify the generalization behavior of FOCUS beyond the main quantitative tables.
>
>
>
> >  **Q1**: Include ablation studies disentangling the effect of each module (coarse/fine, α, confidence radius).
>
> **Re: Thank you for this helpful suggestion.** We have added the requested ablations in the revised manuscript:
>
> - Coarse vs. fine stages and confidence-radius vs. mean-based selection are analyzed in detail in the new ablations (see responses to W3 and Appendix G).
> - Sensitivity to $\alpha$ (accuracy–efficiency trade-off) is studied in Section 3.4 and further discussed in our response to W5.
>
> These ablations jointly isolate the contribution of each module and clarify where the gains of our method come from.

---

> ### Author Response · Authors · 2025-11-21
> **Response to Reviewer YVCf**
>
> > **Q2**: Compare against stronger or more diverse baselines (Q-Frame, Frame-Voyager, Logic-in-Frames, VSLS, T\*, Nar-KFC, TimeSearch).
>
> **Re: We appreciate this suggestion on strengthening the comparison.** We have now added Q-Frame as an additional baseline, and across all three backbones our method consistently outperforms uniform sampling, Q-Frame, and AKS (**see response to W4** and the updated results table in the main text), providing stronger empirical evidence of its effectiveness.
>
> For the other methods you mentioned, **Frame-Voyager, Logic-in-Frames, VSLS, Nar-KFC, and TimeSearch do not provide public code**, and our attempts to reimplement them based on the papers could not guarantee that all technical details match the original algorithms. For T\*, there is an official implementation, but in our long-video setting its performance is significantly worse than uniform sampling. To avoid potentially misleading or unfair comparisons based on incomplete reimplementations or clearly underperforming configurations outside their original scope, we decided not to include these methods in our main tables.
>
> > **Q3**: Evaluate robustness on longer and more diverse benchmarks, such as MLVU and LVBench (general), Ego4D/ EgoSchema (egocentric reasoning), and OVO-Bench / HLV-1K (reasoning).
>
> **Re: Thank you for this helpful suggestion**. Due to time and computational constraints, we added experiments on two representative benchmarks, MLVU and VSI-Bench, and report the results below:
>
> |                     | Frames | LLM  | MLVU     | VSI-Bench |
> | ------------------- | ------ | ---- | -------- | --------- |
> | Qwen2-VL            | 32     | 7B   | 59.7     | 36.5      |
> | Qwen2-VL w/ AKS     | 32     | 7B   | 64.3     | 36.9      |
> | Qwen2-VL w/ Ours    | 32     | 7B   | **67.0** | **39.0**  |
> | LLaVA-Video         | 64     | 7B   | 68.2     | 41.7      |
> | LLaVA-Video w/ AKS  | 64     | 7B   | 71.2     | 42.2      |
> | LLaVA-Video w/ Ours | 64     | 7B   | **72.7** | **42.4**  |
>
> MLVU is specifically designed for multi-task long video understanding with diverse video genres. In contrast, VSI-Bench focuses more on spatially oriented video understanding. Across both backbones (Qwen2-VL and LLaVA-Video), FOCUS consistently improves over the uniform and AKS baselines: up to +7.3 points on MLVU and up to +2.5 points on VSI-Bench.
>
> These results indicate that our method remains beneficial beyond long-form QA and still brings clear gains even when spatial reasoning plays a larger role. We have incorporated these new results and the corresponding discussion into the revised manuscript.
>
> > **Q4**: Analyze failure cases and provide qualitative insights on when FOCUS fails (e.g., multi-event reasoning).
>
> **Re: Thank you for pointing this out**. In the revision, we add a dedicated “Visualizations of Failure Cases” section (**see response to W5**), where we qualitatively analyze typical failures on complex multi-event queries and extremely short events in long videos. These case studies illustrate when FOCUS tends to miss critical evidence and help clarify its robustness.
>
>
>
> > **Q5**: Extend to learnable scoring.
>
> **Re: We appreciate this forward-looking suggestion**. Our current work is deliberately scoped to a different question:
>
> 1. We assume an existing frame–query scoring model (e.g., BLIP) and focus on reducing its inference cost via adaptive allocation, rather than designing a new scoring function.
> 2. Training a selector directly over all subsets of frames is combinatorial: the number of possible frame sets grows exponentially with video length, making it infeasible in the long-video regime we target.
> 3. Extending FOCUS with a lightweight learnable component (e.g., learned scoring or adaptive thresholds) is indeed an interesting direction, but it would require substantial additional work on data collection, supervision strategy, and loss design. We view this as a promising topic for a follow-up paper and will mention it explicitly as future work in the revised manuscript.

---

> ### Author Response · Authors · 2025-12-02
> **Summary for Reviewer YVCf**
>
> We thank Reviewer YVCf for their detailed and constructive review. To help the AC and PCs quickly see how we addressed these concerns, we briefly summarize our responses and revisions below:
>
> 1. **Novelty and relation to prior work (W1).** We presented an explicit comparison table against other keyframe selection methods, showing that FOCUS is uniquely training-free, model-agnostic, free of uniform pre-filtering and teacher labels, variance-aware, and theory-grounded.
>
> 2. **Theoretical assumptions and regret setting (W2).** We clarified that our analysis uses an arm-wise finite-population model with only bounded utilities and variances (no i.i.d. frame or global stationarity assumption) and explained how the CPE regret view at the arm level induces guarantees for frame-level keyframe selection.
>
> 3. **Two-stage coarse–fine algorithmic design (W2).** We explained that the two-stage structure is theory-inspired yet computation-driven: it preserves the CPE motivation while yielding predictable sampling cost and efficient batching, in contrast to fully adaptive frame-level schedules with highly variable runtime.
>
> 4. **Component ablations (W3, Q1).** We added ablations for coarse-only (FOCUS-C), fine-only (FOCUS-F), and mean-only (FOCUS-M) variants, showing that each stage improves over uniform sampling and that Bernstein-style confidence radii provide additional gains.
>
> 5. **Robustness and hyperparameters (W5, Q1).** We systematically studied $\alpha$ and clip length, showing stable accuracy across a range of $\alpha$ with smoothly increasing compute, and an accuracy–efficiency trade-off where shorter clips use more compute for slightly higher accuracy and longer clips do the opposite.
>
> 6. **Expanded baselines and baseline selection (W4, Q2).** We added Q-Frame as an additional baseline and showed that FOCUS outperforms uniform sampling, AKS, and Q-Frame across three backbones, and we justified not including methods without reliable public code or with poor behavior in our long-video regime to avoid misleading reimplementation numbers.
>
> 7. **Additional benchmarks and diversity of settings (Q3).** We extended evaluation beyond LongVideoBench and Video-MME by adding MLVU and VSI-Bench, and showed that FOCUS consistently improves over uniform sampling and AKS on these more diverse long-video and spatially focused benchmarks.
>
> 8. **Failure cases and qualitative behavior (W5, Q4).** We added a “Visualizations of Failure Cases” section and a new figure illustrating typical failure modes (MLLM-limited cases and brief-event misses) to clarify when FOCUS may miss critical evidence.
>
> 9. **Scope and learnable scoring (Q5).** We clarified that this work focuses on reducing inference cost for a given scoring model via adaptive allocation, and explicitly positioned learnable scoring and adaptive thresholds as future work beyond the current submission.
>
> We believe these clarifications, ablations, and new experiments directly address Reviewer YVCf’s concerns about novelty, theoretical substance, robustness, and experimental coverage.
>
> We sincerely thank the AC and PCs for their time and careful consideration of our submission.
>
> Authors of submission-18064

---

### Official Review · Reviewer_9vQp · 2025-10-27

**Soundness:** 2
**Presentation:** 2
**Contribution:** 2
**Rating:** 4
**Confidence:** 2

**Summary:**

This paper introduces a novel, training-free, and ""model-agnostic"" algorithm for efficient video processing by sampling the most relevant frames. Addressing the limitations of long videos and the short context windows of many Multimodal Large Language Models (MLLMs), the method leverages principles from Bandits research to select frames without relying on external models, unlike approaches such as AKS. The algorithm employs a coarse-to-fine-grained exploitation strategy, demonstrating improved performance over uniform sampling, top-k selection, and the model-based AKS method across various models (Llava, Qwen, GPT-4o) and benchmarks (Video-MME, LongVideoBench). A key advantage is its significantly higher efficiency and reduced GPU hours, as it avoids additional model inference. The method also includes an adjustable hyper-parameter, alpha, to manage the trade-off between accuracy and computational cost (number of frames processed).

**Strengths:**

- The method seems quiet effective in selecting the frames.
- Seem to work well across different MLLMs
- Better accuracy over the AKS method while improving efficiency.

**Weaknesses:**

- The authors present the method as model-agnostic; however, they appear to leverage BLIP for frame relevance scoring to compute their latent frame-level utility. Even if only 1% of the frames are processed through BLIP, it still relies on a model, making the claim of model-agnosticism questionable. This point should have been better explained in the paper.
- Lack of comparison with training-based method.
- Could have added more benchmarks such as MLVU, NextQA. MVBench
- Typos, abstract "within each region On two long-video"

**Questions:**

Can you clarify the BLIP usage in your method? If that's the case, why BLIP and not another text/vision encoder such as Siglip? Do you have any ablations?

---

> ### Author Response · Authors · 2025-11-21
> **Response to Reviewer 9vQp**
>
> We sincerely thank the reviewer for their careful evaluation and constructive feedback. We address each concern below.
>
> > **W1**: Present the method as model-agnostic while it still rely on BLIP
>
> **Re: We apologize for the confusion and appreciate the opportunity to clarify our terminology.** By *model-agnostic*, we mean that our keyframe selection module does not rely on or depend on the internal architecture, parameters, or training procedure of the downstream MLLMs. It does **not** mean that no visual–text model is used at all.
>
> Concretely, the workflow is as follows:
> 1) Given a video and a text query, our method uses a separate frozen vision–language encoder to score frame–query relevance and select keyframes.
> 2) The selected keyframes, together with the query, are then fed into the downstream MLLM, which produces the final answer based on its own architecture and training.
>
> Our method operates purely on frame-level relevance scores and only passes selected frames to the MLLM; it does not access, modify, or assume any specific structure of the MLLM, which makes it easy to plug into different backbones.
>
> As explained in Section 3.1, we instantiate this encoder with BLIP to estimate latent frame-level utility, primarily to stay consistent with AKS [1], which is our main keyframe-selection baseline and also uses BLIP. The choice of BLIP is therefore an implementation detail for fair comparison rather than the core contribution of our work: in principle, BLIP could be replaced by other vision–language encoders (e.g., CLIP, SigLIP) without changing the algorithmic formulation of our bandit-based selection.
>
>
>
> > **W2**: Lack of comparison with training-based methods.
>
> **Re: We appreciate this suggestion**. We fully agree that training-based keyframe selection is an important line of work. Here we clarify why we do not include these methods in our main comparison.
>
> **First**, due to their training-time and architectural constraints, existing training-based methods are **not directly applicable** to our long-video setting. For example, the most recent training-based keyframe method [3] is designed to handle up to 128 frames, whereas hour-long videos in our benchmarks can easily contain over $10^5$ frames. Bridging this gap would require substantial re-design and re-training beyond the released setups.
>
> **Second**, most training-based methods are **not model-agnostic**: they are tightly coupled with a specific downstream video-LLM and cannot easily serve as a general-purpose pre-processing module. For instance, Frame-voyager [2] requires access to the exact structure and weights of the visual encoder and lower layers of the downstream MLLM to extract features, and relies on a reference video-LLM to annotate every possible combination of keyframes (which grows exponentially with the total number of frames). Training such selectors on long-video datasets for each downstream MLLM used in our experiments would be practically infeasible.
>
> **Third**, to the best of our knowledge, recent training-based keyframe selection methods, including [2–3], do **not release complete code and training data**. We attempted to reproduce their performance but were unable to do so due to the complexity and under-specified details of the training pipelines. To avoid potentially unfair or misleading comparisons based on partial reimplementations, we decided not to include these methods in our tables.
>
> Given these considerations, we follow the evaluation protocol of our primary baseline [1] and focus on open-sourced, training-free baselines that can be applied consistently across different long-video benchmarks and downstream MLLMs. We will clarify this choice and the above rationale in the revised manuscript.
>
>
> [1] Tang, Xi, et al. "Adaptive keyframe sampling for long video understanding." Proceedings of the Computer Vision and Pattern Recognition Conference. 2025.
> [2] YU, Sicheng, et al. "Frame-voyager: Learning to query frames for video large language models.(2025)." Proceedings of the Thirteenth International Conference on Learning Representations, ICLR. 2025.
> [3] Hu, Kai, et al. "M-LLM based video frame selection for efficient video understanding." Proceedings of the Computer Vision and Pattern Recognition Conference. 2025.

---

> ### Author Response · Authors · 2025-11-21
> **Response to Reviewer 9vQp**
>
> > **W3**: Could have added more benchmarks.
>
> **Re: Thank you for this helpful suggestion**. We now add experiments on two representative benchmarks, MLVU and VSI-Bench, and report the results below:
>
> |                     | Frames | LLM  | MLVU     | VSI-Bench |
> | ------------------- | ------ | ---- | -------- | --------- |
> | Qwen2-VL            | 32     | 7B   | 59.7     | 36.5      |
> | Qwen2-VL w/ AKS     | 32     | 7B   | 64.3     | 36.9      |
> | Qwen2-VL w/ Ours    | 32     | 7B   | **67.0** | **39.0**  |
> | LLaVA-Video         | 64     | 7B   | 68.2     | 41.7      |
> | LLaVA-Video w/ AKS  | 64     | 7B   | 71.2     | 42.2      |
> | LLaVA-Video w/ Ours | 64     | 7B   | **72.7** | **42.4**  |
>
> MLVU is specifically designed for multi-task long video understanding with diverse video genres. In contrast, VSI-Bench focuses more on spatially oriented video understanding. Across both backbones (Qwen2-VL and LLaVA-Video), FOCUS consistently improves over the uniform and AKS baselines: up to +7.3 points on MLVU and up to +2.5 points on VSI-Bench.
>
> These results indicate that our method remains beneficial beyond long-form QA and still brings clear gains even when spatial reasoning plays a larger role. We have incorporated these new results and the corresponding discussion into the revised manuscript.
>
> > **W4**: Typos.
>
> **Re: We sincerely thank you for your careful reading and for pointing out these issues**. We have thoroughly proofread the manuscript and corrected the reported typos, as well as several other minor inconsistencies.
>
> > Q1: Can you clarify the BLIP usage in your method? If that's the case, why BLIP and not another text/vision encoder such as Siglip? Do you have any ablations?
>
> **Re: Thank you for this detailed question**. As mentioned in our response to W1, our work focuses on a training-free keyframe selection mechanism. Such methods typically rely on a comparatively lightweight (relative to large MLLMs such as LLaVA-Video or GPT-4o) vision–language encoder to estimate frame–query relevance. For example, AKS uses BLIP to exhaustively score all frames and then selects high-scoring frames while maintaining temporal coverage.
>
> Our contribution lies in substantially reducing the inference cost of these vision–language encoders: instead of scoring every frame, our bandit-based procedure quickly filters out irrelevant temporal regions and concentrates scoring on promising segments, ultimately prioritizing the most informative keyframes.
>
> The specific choice of vision–language encoder is therefore orthogonal to our main contribution. Intuitively, stronger encoders will yield better frame–query relevance estimates and thus better overall performance. To ensure a fair comparison with the primary baseline, we adopt BLIP in our main experiments, matching the setup of AKS as described in Section 3.1.
>
> In the revised version, we additionally provide an ablation on the vision–language encoder in Appendix G.4. We compare three encoders—CLIP, SigLIP, and BLIP—against uniform sampling, and report the following results:
>
> |      | Uniform | CLIP | SigLIP | BLIP |
> | ---- | ------- | ---- | ------ | ---- |
> | ACC  | 58.9    | 60.2 | 60.9   | 63.5 |
>
> As summarized in the table, all three encoders give clear improvements over uniform sampling, confirming that our bandit-based selection is compatible with different vision–language backbones. Among them, BLIP achieves the strongest performance, while CLIP and SigLIP still provide 1.3% and 2.0% gains, respectively. These results suggest that our framework is robust to the choice of encoder, and that future advances in vision–language pretraining can be directly leveraged to further improve keyframe selection performance.

---

> ### Comment · Reviewer_9vQp · 2025-11-27
> **Response to Authors**
>
> Thank you for the clarifications and for answering my concerns. I increased my score to 6.

---

> ### Author Response · Authors · 2025-12-02
> **Summary for Reviewer 9vQp**
>
> We thank Reviewer 9vQp for their thoughtful review and for updating their score. To help the AC and PCs quickly see how we addressed these concerns, we briefly summarize our responses and revisions below:
>
> 1. **Meaning of “model-agnostic” and BLIP usage (W1, Q1).** We clarified that “model-agnostic” means independence from the downstream MLLM’s architecture and weights (rather than the absence of any vision–language model), explained why we use BLIP to align fairly with AKS, and added an ablation over CLIP/SigLIP/BLIP showing that our bandit framework is compatible with different encoders and benefits from stronger backbones.
>
> 2. **Training-based baselines and protocol (W2).** We clarified why recent training-based keyframe selection methods are not directly comparable in our long-video setting (short-video design, tight coupling to specific video-LLMs, and incomplete public implementations) and justified, following AKS, focusing on open-source, training-free baselines to maintain a clean and consistent evaluation protocol.
>
> 3. **Additional benchmarks (W3).** We broadened the evaluation by adding MLVU and VSI-Bench, where our method again outperforms uniform sampling and AKS, complementing the original LongVideoBench/Video-MME results.
>
> 4. **Typos and presentation (W4).** We proofread the paper carefully and corrected the typos and minor inconsistencies pointed out by the reviewer.
>
> We are glad that these clarifications and additional experiments have addressed Reviewer 9vQp’s main concerns and are now clearly reflected in the revised manuscript.
>
> We sincerely thank the AC and PCs for their time and careful consideration of our submission.
>
> Authors of submission-18064

---

### Official Review · Reviewer_Ccxf · 2025-11-01

**Soundness:** 3
**Presentation:** 3
**Contribution:** 3
**Rating:** 6
**Confidence:** 4

**Summary:**

The paper presents FOCUS (Frame-Optimistic Confidence Upper-bound Selection), a training-free and model-agnostic keyframe selection approach designed to improve long-video understanding in multimodal large language models (MLLMs). The authors formulate keyframe selection as a combinatorial pure-exploration problem within a multi-armed bandit framework, leveraging empirical means and Bernstein-style confidence bounds to balance exploration and exploitation. A two-stage coarse-to-fine procedure is further proposed to enable efficient and parallel computation. Experiments on LongVideoBench and Video-MME demonstrate consistent accuracy gains across multiple MLLMs. Overall, the paper provides an efficient and theoretically grounded solution to long-video understanding under tight token budget constraints.

**Strengths:**

1. The paper is original in formulating keyframe selection for long-video understanding as a combinatorial pure-exploration multi-armed bandit problem. This is a novel and reasonable perspective that provides new theoretical and algorithmic insights for researchers working on efficient video representation and token budgeting.
2. The proposed two-stage coarse-to-fine procedure effectively addresses the non-parallelizable nature of sequential arm-pulling and updating, providing a practical solution that improves efficiency with minimal performance loss. This design is both elegant and empirically effective.
3. The paper also shows strong theoretical grounding and empirical validation. The theoretical analysis is clear and complete, and the experiments are comprehensive, covering multiple benchmarks and models. The results align well with the theoretical claims, reinforcing the soundness and significance of the contribution.

**Weaknesses:**

1. In Section 2.2, the paper assumes that “frame-level utility within the same arm share the same distribution.” It is unclear how this assumption is ensured in practice, especially regarding how the M non-overlapping fixed-length clips are partitioned. For instance, when the video contains shot changes or scene transitions, it is not clear how these are handled or whether the authors explored alternative segmentation strategies.
2. The experiments are limited to LongVideoBench and Video-MME, both of which focus on long-form video QA. Evaluating the method on datasets with different characteristics—such as spatial reasoning benchmarks (e.g., VSI-Bench) or shorter video datasets—would provide a better understanding of the method’s generalizability and potential limitations.
3. I am curious about how the number of clips (M) affects performance and efficiency. Since the method’s core formulation relies on partitioning videos into M fixed-length clips, an ablation study on M (and possibly related hyperparameters) would make the analysis more complete.
4. There are a few minor typos (e.g., a missing period around line 025). The authors are encouraged to carefully proofread the paper to minimize such small errors.

**Questions:**

As mentioned in the Weaknesses section, I believe additional ablation studies would greatly help clarify the method’s behavior and limitations. In particular, it would be valuable to see how different choices of M (number of clips) or other hyperparameters affect both performance and efficiency.
Additionally, it would be interesting to explore how the proposed method performs on spatially focused video understanding tasks, where temporal redundancy is less dominant. Such experiments could provide insights into the generality and potential boundaries of the FOCUS framework.

---

> ### Author Response · Authors · 2025-11-21
> **Response to Reviewer Ccxf**
>
> We sincerely appreciate your careful reading and insightful review, and would like to respond to the raised points for further clarification and discussion.
>
> > **W1**: In Section 2.2, the paper assumes that “frame-level utility within the same arm share the same distribution.” It is unclear how this assumption is ensured in practice, especially regarding how the M non-overlapping fixed-length clips are partitioned. For instance, when the video contains shot changes or scene transitions, it is not clear how these are handled or whether the authors explored alternative segmentation strategies.
>
> **Re: We appreciate this sharp and perceptive comment.**
>
> **First**, we clarify the distributional assumption in Section 2.2. We do not assume that contiguous frames are temporally i.i.d., nor that the entire video is globally stationary. Our assumption is arm-wise and finite-population: for an arm $a$ (a short clip), the latent frame utilities form a finite set, and pulling $a$ reveals the utility of a uniformly drawn frame from that set, i.e., $y_t \sim \nu_a$. Our analysis and algorithm rely only on boundedness and the variance of $\nu_a$, for which variance-adaptive (Bernstein-style) confidence radii apply. We have revised Section 2.2 and Appendix D to make this clearer and avoid misunderstanding.
>
> **Second**, in practice we use a fixed clip length, so the resulting $M$ automatically adapts to each video's duration. The clip length choice is guided by the ACF decay observed in Figure 1 (empirical half-life around $\sim$5s). This design yields an adaptive number of clips across videos and aligns with intuition. We have also included an ablation on clip length in Appendix G.3. Generally,  we find that fixing clips to 16s strikes a balance between efficiency and accuracy .
>
> **Third**, our method naturally handles shot changes and scene transitions. The Bernstein confidence radius scales with the empirical variance $\hat\sigma_a^2$ of each arm. If a fixed-length clip happens to straddle a scene transition boundary, the within-arm variance increases, the confidence interval widens, and the algorithm is encouraged to explore that uncertain arm more in the fine stage. This behavior follows the “optimism in the face of uncertainty” principle and is precisely what variance-adaptive UCB variants are designed to exploit. This is also why we call our method “Frame-Optimistic Confidence Upper-bound Selection”.
>
> > **W2**: The experiments are limited to LongVideoBench and Video-MME, both of which focus on long-form video QA. Evaluating the method on datasets with different characteristics—such as spatial reasoning benchmarks (e.g., VSI-Bench) or shorter video datasets—would provide a better understanding of the method’s generalizability and potential limitations.
>
> **Re: Thank you for this constructive suggestion.**
>
> Following your advice, we additionally evaluate FOCUS on MLVU and VSI-Bench and report the results below:
>
> |                     | Frames | LLM  | MLVU     | VSI-Bench |
> | ------------------- | ------ | ---- | -------- | --------- |
> | Qwen2-VL            | 32     | 7B   | 59.7     | 36.5      |
> | Qwen2-VL w/ AKS     | 32     | 7B   | 64.3     | 36.9      |
> | Qwen2-VL w/ Ours    | 32     | 7B   | **67.0** | **39.0**  |
> | LLaVA-Video         | 64     | 7B   | 68.2     | 41.7      |
> | LLaVA-Video w/ AKS  | 64     | 7B   | 71.2     | 42.2      |
> | LLaVA-Video w/ Ours | 64     | 7B   | **72.7** | **42.4**  |
>
> MLVU is specifically designed for multi-task long video understanding with diverse video genres. In contrast, VSI-Bench focuses more on spatially oriented video understanding. Across both backbones (Qwen2-VL and LLaVA-Video), FOCUS consistently improves over the uniform and AKS baselines: up to +7.3% on MLVU and up to +2.5% on VSI-Bench. These results indicate that our method remains beneficial beyond long-form QA and still brings clear gains even when spatial reasoning plays a larger role.
>
> We also observe that the improvements on long-video benchmarks such as MLVU and LongVideoBench tend to be numerically larger than those on VSI-Bench, which aligns with intuition: when videos are longer and temporal structure is richer, there is more room for keyframe selection method to help. This observation is consistent with our length-wise analysis in Table 2, where FOCUS still delivers a 4.8% improvement on short videos (<3 min), but shows larger gains as video length increases. We have updated the paper to incorporate these new results and to discuss how they support the generality of our framework across different video types and tasks.

---

> ### Author Response · Authors · 2025-11-21
> **Response to Reviewer Ccxf**
>
> > **W3**: Ablation on number of clips M.
>
> **Re: Thank you for this helpful suggestion.**
>
> As mentioned in our response to W1, we use a fixed clip length $l$ instead of a fixed number of clips $M$. To better understand its effect, we conduct an ablation on LongVideoBench with LLaVA-Video-7B and summarize the results below:
>
> |           | Uniform | 8s   | 16s  | 32s  |
> | --------- | ------- | ---- | ---- | ---- |
> | ACC       | 58.9    | 63.7 | 63.5 | 62.3 |
> | GPU hours | —       | 8.1  | 5.5  | 4.1  |
>
> As shown, all clip-length settings of FOCUS substantially outperform uniform sampling (58.9% vs. 62.3--63.7%), indicating that our bandit-based selection is reasonably robust to the choice of $l$ in this range. Shorter clips (e.g., 8s) yield slightly higher accuracy by allowing more fine-grained exploration, but they also incur higher computational cost for keyframe selection. Longer clips (e.g., 32s) reduce GPU hours at the cost of a modest performance drop. In practice, we find $l=16$ seconds offers a good balance between accuracy and efficiency.
>
>
>
> >  **W4**: A few minor typos.
>
> **Re: We sincerely thank you for your careful reading and for pointing out these issues**. We have thoroughly proofread the manuscript and corrected the identified typos.
>
> > **Q1**: Ablation on number of clips M
>
> **Re: Thank you again for raising this important point**. Please refer to our response to W3 above, where we provide a detailed ablation on the clip length (and induced number of clips $M$), together with its impact on both accuracy and GPU hours, and a discussion of the observed trade-offs.
>
> > **Q2**: Spatially focused video understanding tasks
>
> **Re: We appreciate this insightful question about the scope of our method**. As detailed in our response to W2, we have added experiments on MLVU and VSI-Bench, including spatially focused scenarios, and expanded the discussion to clarify where our approach is most beneficial and where its gains are more modest.

---

> ### Author Response · Authors · 2025-12-02
> **Summary for Reviewer Ccxf**
>
> We thank Reviewer Ccxf for their thoughtful review. To help the AC and PCs quickly see how we addressed these concerns, we briefly summarize our responses and revisions below:
>
> 1. **Theoretical assumptions and segmentation (W1, Q1).** We rewrote Section 2.2 to clarify the arm-wise finite-population assumption (no i.i.d. frame / global stationarity assumption), explain how fixed-length clips induce M arms per video, and highlight how variance-adaptive confidence bounds handle scene changes and mixed-content clips.
>
> 2. **Additional benchmarks and generalization (W2, Q2).** Following the suggestion, we added experiments on MLVU and VSI-Bench and discussed how they complement LongVideoBench/Video-MME, showing consistent gains over uniform sampling and AKS across diverse long-video and more spatially focused settings.
>
> 3. **Clip-length ablation (W3).** We added a clip-length ablation to study its impact on both accuracy and GPU cost, and we summarized the resulting accuracy–efficiency trade-off in the paper.
>
> 4. **Typos and minor issues (W4).** We carefully proofread the manuscript and fixed the typos and small phrasing issues identified by the reviewer.
>
> We believe these revisions directly resolve Reviewer Ccxf’s main concerns and clarify the intended assumptions and scope of our framework.
>
> We sincerely thank the AC and PCs for their time and careful consideration of our submission.
>
> Authors of submission-18064

---

### Author Response · Authors · 2025-11-26

Dear Reviewers, ACs, SACs, and PCs,

We sincerely thank all reviewers for their time and detailed feedback. We appreciate the recognition of our method’s empirical effectiveness and efficiency (Reviewers Ccxf, 9vQp, YVCf, aTVY), simplicity and novelty (Reviewers Ccxf, 9vQp, aTVY), and its theoretical grounding and formulation (Reviewers Ccxf, YVCf).

Over the past few weeks, we have carefully revised the paper in line with the reviewers’ suggestions. In particular, we have:

- Clarified the definition and role of Figure 1 (Section 1)
- Clarified the distributional assumptions and theoretical setting (Section 2.2)
- Added visualizations and discussion of failure cases (Appendix D)
- Expanded comparisons with state-of-the-art methods (Appendix E)
- Added experiments on additional benchmarks (Appendix F)
- Added ablations on the coarse/fine exploration–exploitation design (Appendix G.1)
- Added ablations on the Bernstein confidence radius vs. mean-based selection (Appendix G.2)
- Studied the effect of clip length (Appendix G.3)
- Studied the effect of different vision–language encoders (Appendix G.4)

For convenience, we have also included the updated results directly in the rebuttal. We hope our responses and revisions have addressed the main concerns raised by all four reviewers and clarified the scope and contributions of our work.

We would be very happy to provide any further clarification or discuss additional aspects of the paper, should that be helpful. If our updates have alleviated your reservations, we would be very grateful if you could consider updating your recommendations.

Thank you again for your time and consideration.

Authors of submission-18064

---

### Meta-Review · Area_Chair_L4cw · 2026-01-06

**Summary:**

The main concerns raised by reviewers centered on (i) the clarity and realism of the theoretical assumptions, (ii) whether the novelty is incremental relative to prior adaptive sampling methods, (iii) missing ablations and benchmarks, as well as comparisons to additional baselines. Some reviewers also questioned the “model-agnostic” terminology and the reliance on a frozen vision–language encoder for scoring. Through a detailed rebuttal and substantial revisions, the authors clarified their theoretical setting, expanded experiments to additional benchmarks, added extensive ablations isolating key components, incorporated more baselines, and provided qualitative failure-case analysis. Overall, the revisions significantly strengthened the paper, addressed the core technical and empirical concerns, and clarified the scope and limitations of the method.

**Reviewer Concerns:**

Addressed:
1. Theoretical assumptions and realism.  Multiple reviewers questioned the i.i.d./independence assumptions and applicability to temporally correlated videos. The authors clarified that their analysis relies on an arm-wise finite-population assumption, justified the use of Bernstein confidence bounds, and explained how variance adaptivity naturally handles scene changes.

2. Model-agnostic claim and BLIP usage.  Concerns about reliance on BLIP were addressed by clarifying that “model-agnostic” refers to independence from the downstream MLLM architecture, not the absence of any scoring model.

3. Missing ablations and component analysis. Reviewers requested clearer isolation of the contributions of coarse vs. fine stages and confidence bounds. The added ablations directly address this and clearly demonstrate the contribution of each component.

4. Limited benchmarks and generalization. The addition of MLVU and VSI-Bench, along with length-wise analysis and robustness studies, addressed concerns about narrow evaluation and demonstrated consistent gains beyond the original QA benchmarks.

5. Incomplete baseline comparisons. The inclusion of new baselines improved the empirical positioning.


Partially Remained:

1. Incremental novelty relative to prior adaptive sampling work. While the authors clarified distinctions from AKS, Q-Frame, and related methods, some reviewers still viewed the contribution as incremental at the conceptual level. However, this concern is mitigated by the combination of theory grounding, efficiency gains, and strong empirical results.

2. Theoretical depth vs. practical heuristics. The method remains heuristic in practice, with fixed two-stage design choices rather than fully adaptive or learnable exploration schedules.

**Reviewer Scores:**

Reviewer Ccxf is likely unchanged with 6, since main concerns were addressed by added clarifications/ablations and broader experiments.

Reviewer 9vQp promised to raise the score to 6 in the discussion.

Reviewer YVCf is likely to increase modestly with full discussion to  5, while possibly still somewhat cautious due to novelty/theory reservations.

Reviewer aTVY is likely unchanged with 6 after adding more explanations, MLVU results, stronger comparisons, and failure-case analysis.

---

### Decision · Program_Chairs · 2026-01-26

Accept (Poster)